



# Diurnal aging of biomass burning emissions: Impacts on secondary organic aerosol formation and oxidative potential

Maria P. Georgopoulou[1,2], Kalliopi Florou[1], Angeliki Matrali[1,2], Georgia Starida[2], Christos Kaltsonoudis[1], Athanasios Nenes[1,3,*], and Spyros N. Pandis[1,2,*]

[1]Institute of Chemical Engineering Sciences, Foundation for Research and Technology Hellas (FORTH/ICE-HT), Patras 26504, Greece

[2]Department of Chemical Engineering, University of Patras, Patras 26504, Greece

[3] Laboratory of Atmospheric Processes and their Impacts, School of Architecture, Civil & Environmental Engineering École Polytechnique Fédérale de Lausanne CH-1015 Lausanne, Switzerland

*Correspondence to*: Spyros Pandis (spyros@chemeng.upatras.gr) and Athanasios Nenes (athanasios.nenes@epfl.ch)

## 1  Abstract

Residential biomass burning is an important wintertime source of aerosols. These particles are subjected to complex diurnal aging processes in the atmosphere, contributing to urban and regional air pollution. The cumulative impact of these aging cycles on aerosol composition and oxidative potential, a key toxicity metric, remains unclear. This study examined the oxidation cycles of biomass burning emissions during day-to-night and night-to-day transitions in the FORTH (Foundation for Research and Technology – Hellas) atmospheric simulation chamber, focusing on emissions from burning of olive wood. The final high-resolution AMS spectra of biomass burning organic aerosol (bbOA) after either oxidation cycle were almost identical ($R^2$ > 0.99, $\theta = 3°$). This indicates transformation into similar biomass burning secondary organic aerosol (bbSOA) regardless of the initial step of the diurnal cycle. A 56% average increase in the bbOA oxygen-to-carbon (O:C) ratio was observed during both cycle cases (from 0.38 ± 0.06 for the fresh to 0.59 ± 0.07 after aging). Additional OA mass was produced after the two cycles, varying from 35 to 90 % of the initial OA. The aging of the emissions led to a final water-soluble oxidative potential (WS-OP) increase of 60% to 68 ± 18 pmol min$^{-1}$ µg$^{-1}$for both cycles, but with notably different transient values that depend on the order of the oxidation regimes. The effect of each oxidation regime on the WS-OP of the bbOA depends on the airmass history. The evolution of the WS-OP was not well correlated with that of the O:C.



## 1 Introduction

Biomass burning for residential heating has significantly increased over the past two decades in several countries, primarily driven by rising energy costs and efforts to reduce the use of fossil fuels (Alper et al., 2020). Alongside contributions from wildfires, residential biomass burning has emerged as a major source of urban and regional pollution worldwide (Zauqi-Sajani et al., 2024). Solid biomass currently represents nearly 45% of the total bioenergy supply in the EU, 40% of which is allocated to residential heating, with an anticipated 20% increase projected by 2050 (IEA, 2019, 2021; Reid et al., 2020). This upward trend in the residential burning of solid biomass, particularly wood, has raised serious concerns regarding air quality and human health (Cincinelli et al., 2019; Guercio et al., 2021; Pardo et al., 2024).

Particles emitted from biomass burning consist of organic compounds, elemental carbon (EC), sulfates, nitrates, ammonium, and ash (Jiang et al., 2024). Biomass burning emissions also include a range of gases; carbon monoxide (Shen et al., 2020), volatile organic compounds (VOCs) such as aldehydes, ketones, and organic acids (Zhang et al., 2021; Huang et al., 2022), carcinogenic polycyclic aromatic hydrocarbons (PAHs and oxy-PAHs) (Tsiodra et al., 2021, 2024; Lim et al., 2022), as well as nitrogen oxides and ammonia (Bray et al., 2021). The emitted VOCs contribute to the formation of biomass burning secondary organic aerosol (bbSOA) and can have direct health effects (Fang et al., 2021). The emission profile of these pollutants is variable, influenced by factors such as fuel type and quality (e.g., logs vs. pellets; hardwood vs. softwood; certified vs. non-certified wood, moisture content etc.), burning conditions (e.g., flaming vs. smoldering, air/oxygen supply, and dilution), and the type of combustion appliance (Fachinger et al., 2017; Nyström et al., 2017; Price-Allison et al., 2021; Trubetskaya et al., 2021).

After their release, biomass burning emissions are subject to chemical transformations through homogeneous or heterogeneous reactions, that differ between daytime and nighttime (Donahue et al., 2012; Hodshire et al., 2019; Yazdani et al., 2023). During these reactions, a significant amount of SOA (Yazdani et al., 2023) and reactive oxygen species (ROS) (Wang et al., 2023) can be generated. Hennigan et al. (2011) reported significant variability in bbSOA formation during the photo-oxidation of different emissions. Yazdani et al. (2023) reported that after 6 to 10 hours of daytime exposure, up to 30% (with an average of 15%) of the primary bbOA (bbPOA) mass was oxidized, forming bbSOA that was predominantly composed of acids. The coupled gas-particle partitioning, and reaction of semi-volatile vapors (SVOCs) may play an important role in the processing of bbPOA (Hennigan et al., 2011; Srivastava et al.,





2022). Li et al. (2024) demonstrated that intermediate volatility species (IVOCs) can contribute
approximately 70% of the formed bbSOA, more than twice the contribution from VOCs.

The nighttime oxidation of biomass burning emissions by the nitrate radical ($NO_3$) also

leads to rapid aerosol changes (Kodros et al., 2020), but to a lesser extent compared to OH
oxidation (Yazdani et al., 2023). In some cases, a doubling of bbOA levels compared to the
initial primary bbOA has been observed. This increase has been attributed to gas-phase
reactions between the $NO_3$ radical and mainly phenolic compounds or furanic aldehydes
(Hartikainen et al., 2018). Moreover, a substantial increase (7-100%) in the aerosol oxygen-to-
carbon (O:C) ratio, as well as in the mass of organic nitrates in bbOA has been reported, as
result of nocturnal aging (Kiendler-Scharr et al., 2016; Kodros et al., 2022; Yazdani et al.,

2023).

To date, field and atmospheric simulation chamber studies have focused on the oxidation

of biomass burning emissions during either daytime or nighttime oxidation regimes, driven
respectively by OH and $NO_3$ radicals (Hennigan et al., 2011; Fry et al., 2014; Hodshire et al.,
2019; Jorga et al., 2021; Kodros et al., 2022; Wang et al., 2023; Yazdani et al., 2023). While
such investigations have significantly advanced our understanding of the individual effects of
these oxidation regimes, they do not fully capture the real-world evolution of biomass burning
aerosols, which undergo multiple repeated cycles of daytime and nighttime chemistry during
their atmospheric lifetime. Studies on successive aging from daytime and nighttime cycling do
exist but have focused on the changes of the optical and chemical properties of bbOA and the
gas-particle phase partitioning of semi- and intermediate-volatility organic compounds (Tiitta
et al., 2016; Hartikainen et al., 2018; Cappa et al., 2020; Che et al., 2022; Desservettaz et al.,
2023; Yazdani et al., 2023). These alternating oxidation regimes cause successive changes in
chemical composition, reactivity, and toxicity (Li et al., 2021, 2023; Tomlin et al., 2022; He et
al., 2024) that are not well understood. Consequently, the timing of atmospheric BB emissions,
being released during the day or night, may also influence the chemical trajectory of BB aerosol
aging and therefore affect its composition and properties, including toxicity.

Biomass burning particles are significant sources of reactive oxygen species (ROS),

including free radicals (e.g., OH, $RO_2$, $HO_2$) and non-radicals (e.g., $^1O_2$, $H_2O_2$). Upon
inhalation, these species interact with biological tissues and can disrupt cellular redox balance,
triggering (or propagating) oxidative stress and systemic health effects (Costabile et al., 2023).
The ability of particulate matter (PM) to catalyze ROS production, known as oxidative
potential (OP), is a critical metric linking aerosol exposure to health outcomes (Zhang et al.,
2022; Dominutti et al., 2025). Among the various in vivo and in vitro methods developed to



quantify OP (Ng et al., 2019), the abiotic dithiothreitol (DTT) assay is the most well established
one, providing a measure of the water-soluble OP (WS-OP) of aerosols through the depletion
of surrogate DTT in aerosol extracts (Cho et al., 2005). The broad sensitivity of this method to
diverse sources of ROS in aerosols with long lifetimes (Gao et al., 2020; Rao et al., 2020),
along with its optimization over the years (Fang et al., 2015; Puthussery et al., 2020) to provide
more rapid measurements of water-soluble OP (WS-OP), makes it highly suitable for large-
scale studies. Studies using the DTT assay have identified bbOA and SOA as dominant
contributors to DTT activity, accounting respectively for 35% and 30% of total OP in ambient
aerosols in the Southeastern USA (Verma et al., 2015). More recent studies confirm that
biomass burning is a significant source of OP in diverse environments, highlighting the
importance of understanding diurnal variations in OP from biomass burning (Paraskevopoulou
et al., 2019, 2022; Mylonaki et al., 2024).
Photochemical aging during daytime oxidation promotes particle-bound ROS
production, enhancing the OP of the aged aerosols (Li et al., 2021; Wang et al., 2023). For
bbOA, the OP was found to increase by a factor of two ($2.1 \pm 0.9$) after multiple days (68 h) of
atmospheric aging (Wong et al., 2019). This implies that the health impacts of bbOA may
extend far from its sources, as it ages and becomes part of the background aerosol
(Vasilakopoulou et al., 2023; Mylonaki et al., 2024).
While it is well-established that bbOA ages rapidly at night, the effects of its nocturnal
aging on aerosol OP are poorly understood. Moreover, to our knowledge no studies have yet
investigated how the oxidation sequence (day-to-night and night-to-day) affects aerosol
chemical composition, aging trajectory, and toxicity (i.e., evolution of OP). This study aims to
address these knowledge gaps through controlled chamber experiments simulating realistic
diurnal oxidation cycles. In these experiments, fresh biomass burning emissions undergo
sequential aging, either through daytime oxidation followed by nighttime oxidation or the
reverse. By comparing day-to-night and night-to-day sequences, we aim to elucidate the
interplay of oxidation regimes on aerosol chemical evolution and OP, providing novel insights
into the health impacts of diurnally aged biomass burning aerosols.

## 2   Methods

### 2.1   Atmospheric simulation chamber experiments

Emission aging experiments took place at the FORTH-ASC chamber facility at Patras,
Greece. Figure 1 illustrates the setup used for conducting the experiments. Fresh biomass



burning emissions were produced in the combustion facility beneath FORTH-ASC by a residential wood stove, fed with commercially available olive wood logs and branches. This type of hardwood is widely used as a fuel in Greece. The emissions were diluted before their injection into the smog chamber, using a custom-made dilution device that was located at the chamber inlet.

The FORTH-ASC consists of 10 m$^3$ squared Teflon chamber, located inside a 30 m$^3$ reflective room (polished interior aluminium walls), which is temperature-regulated and equipped with ultraviolet lights (Osram, L 36W/73 UV lamps). This setup yields a maximum $NO_2$ photo-dissociation rate coefficient ($J_{NO_2}$) of 0.5 min$^{-1}$ when all lights are on. In this study 1/3 to 2/3 of the ultraviolet lights were used during photooxidation, resulting in a $NO_2$ photo-dissociation rate coefficient ($J_{NO_2}$) of 0.17 to 0.33 min$^{-1}$.

Eight day-to-night (denoted as daytime-first or DN) and eight night-to-day (denoted as nighttime-first or ND) aging experiments were performed under dry (12-24% RH) conditions. Table 1 summarizes the initial aerosol composition and experimental conditions for all the conducted experiments. To investigate the impact of fire starter on biomass burning emissions characteristics, pine kindling mixed with olive logs was used in two of the nighttime-first experiments (ND7, ND8). Pine, which is a softwood, has chemically distinct characteristics compared to olive wood (hardwood) and is used as a kindling material because it burns quickly due to its high resin content.

The smog chamber was first flushed with clean air overnight at a rate of 20 L min$^{-1}$. Approximately 30 min after the combustion ignition in the wood stove, when flaming conditions had been achieved, a fraction of the fresh emissions was diluted with clean air (dilution ratio ranging from 1:5 to 1:10) and was injected into the chamber, which was pre-filled with clean air and regulated to the desired RH level. This resulted in additional dilution (dilution ratio ~1:30) of the emissions. Two high precision mass flow controllers (Bronkhorst EL-FLOW Prestige FG-201CVP), operating at flow rates ranging from 0 to 20 L min$^{-1}$, were used; one to supply clean air to the smog chamber during its filling and cleaning stages, and the other to supply clean air to the dilution system. The initial PM$_1$ concentration achieved in the chamber was 112 ± 56 μg m$^{-3}$ on average (Table 1). The fresh emissions were left to equilibrate and were characterized for about 2 hours. 30-90 ppb of d$_9$-butanol (98%, Cambridge Isotope Laboratories) was also injected in the chamber as a tracer to determine the concentration of OH radicals (Barmet et al., 2012).



Subsequently, in daytime-first experiments, the UV lights were turned on, initiating the
daytime aging of fresh biomass burning emissions by OH radicals for at least 2 hours, without
the addition of further oxidants. This oxidation step was then followed by at least 2 hours of
aging with $NO_3$ radicals under dark conditions. To initiate $NO_3$ radical formation, $NO_2$ and $O_3$
were injected into the chamber at concentrations of 50-150 ppb, and 60-280 ppb, respectively.
During nighttime-first cycling experiments, the same oxidation steps were performed but in
reverse order. In daytime-first experiments, "time zero" was defined as the moment when the
UV lights were turned on, whereas in nighttime-first experiments, it was the point at which $O_3$
was injected.

Particle wall losses were also characterized for each experiment. After the completion of
the two oxidation stages ammonium sulfate (($NH_4)_2SO_4$ ≥99 %, Sigma Aldrich) was injected
into the chamber and its loss rate was monitored for at least 3 hours. The dry seeds were
produced by atomizing a ($NH_4)_2SO_4$ solution using a TSI atomizer (model 3076) and drying
the resulting droplets with a diffusion silica gel dryer (Fig. 1), as described in Wang et al.
(2018).

**2.2   Online instrumentation**

A suite of instrumentation was used for the online characterization of both particle and gas-
phase pollutants (Fig. 1). A scanning mobility particle sizer (SMPS; Classifier) model 3080;
DMA, model 3081, TSI) coupled to a butanol-based condensation particle counter (CPC,
model 3775 high, TSI), was used for the measurement of the number and volume size
distributions (mobility diameter in the range of 13–700 nm) of the aerosol particles. The SMPS
sampled every 3 min with its sheath flow rate set at 3 L $min^{-1}$ and the sample flow rate at 0.6
L $min^{-1}$. A high-resolution time-of-flight aerosol mass spectrometer (HR-ToF-AMS, Aerodyne
Research Inc.), working in V mode with vaporizer temperature set at 600°C and sampling flow
rate of approximately 0.1 L $min^{-1}$, was used for monitoring the time evolution of the non-
refractory organic and inorganic $PM_1$ aerosol composition with time resolution of 3 min.
Aerosol absorption and black carbon (BC) concentration were measured with a seven-
wavelength aethalometer (Magee Scientific, Model AE33-7), sampling at 2 L $min^{-1}$. VOCs
were measured using a proton transfer reaction mass spectrometer (PTR-QMS 500, Ionicon
Analytik), sampling at 0.5 L $min^{-1}$. The drift tube was maintained at 2.3 mbar and operated at
600 V. A detailed explanation of the PTR-MS operational parameters and the calibration
procedure using VOC standards can be found in in Kaltsonoudis et al. (2016). Concentrations
of carbon monoxide (CO) and dioxide ($CO_2$), sulfur dioxide ($SO_2$), ozone ($O_3$) and nitrogen



oxides ($NO_x$) were measured using the corresponding monitors; CO (Teledyne model 300E),
$CO_2$ (Teledyne model T360), (Thermo model 43i-TLE), $O_3$ (Teledyne model 400E), NO and
$NO_2$ (Teledyne model T201). The total sampling flow rate of all monitors was 3.8 L $min^{-1}$.

## 2.3    Online data analysis methodology

The initial combustion conditions in the chamber were characterized by calculating the
modified combustion efficiency (MCE) as the ratio of the carbon dioxide ($CO_2$) to the sum of
$CO_2$ and carbon monoxide (CO) (Yokelson et al., 1996).

The HR-ToF-AMS data were analyzed using the packages SQUIRREL (Sequential Igor

data Retrieval; v1.57) and PIKA (Peak Integration by Key Analysis; v1.16) incorporated in
Igor Pro software (WaveMetrics; version 6.37). The method described in Canagaratna et al.
(2015) was used to estimate of elemental O:C ratio. The AMS collection efficiency (CE) and
the corresponding OA density have been determined using the algorithm proposed by
Kostenidou et al. (2007). This approach combines the volume distributions obtained from the
SMPS and the mass distributions of the main $PM_1$ components from the AMS. The BC
concentration obtained by the aethalometer was also included in the calculation, assuming a
size distribution for BC similar to that of OA.

SMPS measurements were corrected using size-dependent wall loss rate constants,

estimated by monitoring the decline in the mass concentration of $(NH_4)_2SO_4$ particles injected
into the chamber at the end of each experiment. Practically size independent first-order wall
loss rates were observed for particle diameters ranging from 60 to 700 nm. Based on this, the
concentrations of the non-refractory $PM_1$ aerosol species measured by the AMS were corrected
using one experiment-specific, size-independent wall loss rate constant that was $0.15 \pm 0.05$
$h^{-1}$ on average.

The total OA was split into primary (bbPOA) and secondary (bbSOA) following the

approach proposed by Jorga et al. (2020) and applied for bbOA by Kodros et al. (2022). To
quantify the variation between primary and secondary bbOA mass spectra, obtained by the
AMS, the theta angle ($\theta$) was estimated (Kostenidou et al., 2009). This angle represents the
inner product of the two spectra (i.e., fresh and aged one), considered as n-dimensional vectors
(n is to the number of the mass-to-charge (m/z) ratios). Theta angles less than 10° imply high
similarity, while major differences between two compared spectra correspond to $\theta$ values
higher than 25° (Florou et al., 2023). The approach described in Kiendler-Scharr et al. (2016)
was used to quantify the particulate organic nitrate (ON). In the present study the minimum



measured $NO_2^+/NO^+$ ratio in all experiments was 0.04. The corresponding measured ratio for
pure $NH_4NO_3$, determined through calibration, was equal to 0.56.
Prior to each experiment, background VOC levels in the chamber were measured using
the PTR-MS for at least 1 hour. The PTR-MS was unavailable during experiments DN2–DN7.
The PTR-MS measurements of the protonated VOCs were background-corrected and averaged
at the end (over the last 1 h) of the fresh emissions' stabilization period, as well as at the end
(over the last 1 h) of each oxidation step. The final values are summarized in Table S1 of the
supplement, along with a classification of the identified VOCs by chemical structure and
functional groups.
Following the work of Barmet et al. (2012), the average OH radical concentration was
estimated from the decline/reduction in the concentration of the $m/z$ 66 (protonated mass of d9-
butanol). A d9-butanol reaction rate coefficient equal to $3.4 \times 10^{-12}$ $cm^3$ molecule$^{-1}$ s$^{-1}$ (at 295
K) was assumed (Allani et al., 2021).
**2.4  Collection of samples for offline analysis**
To investigate the WS-OP of both fresh and aged BB aerosol, as well as to measure their
organic (OC) and elemental carbon (EC) content, filter samples (Whatman Tissuquartz
2500QAO-UP, 47 mm, 0.45 pose size) were collected for 1 h at the end of the emissions'
equilibration period as well as at the end of each oxidation step. Prior to each experiment, blank
filter samples were also collected. Sampling was conducted using a filter holder coupled with
a $PM_{2.5}$ cyclone positioned at the chamber exit. An external vacuum pump (Becker VT 4.10,
150 Mbar), operating at a flow rate of 16.7 L min$^{-1}$, was used, with its exhaust connected to a
HEPA filter (Whatman 6702-9500). Prior to sampling, the quartz filters were baked at 500 °C
for 10 h and left in the oven overnight, to remove any absorbed organic material. Each filter
was wrapped in prebaked aluminum foil and was kept before and after sampling in sterile
polystyrene petri dishes (50 mm, Pall Laboratory). After sampling all filters were stored at a
temperature of -20 °C, until WS-OP and OC/EC analysis.
Tenax sorbent tubes (stainless steel 3.5 x 1/4 in tubes, filled with Tenax TA, Markes
International) were used to collect VOC samples at specific time intervals. The custom-made
sampling system used included a mass flow controller (Alicat Scientific MC-500SCCM-
D/5M), the sampling tube, and a diaphragm vacuum pump (AIRPO, Model D2028B 12VDC),
operating at a flow of 0.3 L min$^{-1}$ for 1 to 1.5 h, resulting in total collected sample volumes
ranging from 18-27 L. After sampling all sorbent tubes were capped with long-term storage



brass caps containing PTFE ferrules and were stored in a freezer at -18 °C (Harshman et al.,
2016).

## 2.5    TD-GCMS measurements

The offline determination of VOCs/IVOCs involved a two-step desorption process. The
compounds adsorbed in the Tenax tubes were first desorbed using a thermal desorber (UNITY–
Air Server-xr, Markes International Ltd.). During thermal desorption (TD), the sorbent tube
underwent heating up to 280 °C for 10 min to release all its contents. Subsequently, the
desorbed VOCs were captured using Helium (as the carrier gas) and then deposited onto a
sorption cold trap at 20 °C. Subsequently, the temperature of the cold trap was gradually
increased from 20 °C to 300 °C at a rate of 100 °C $s^{-1}$, where it remained for 6 min. The retained
analytes were then injected into a single quadrupole gas chromatograph-mass spectrometer
(GSMS, Shimadzu model QP2010, with helium as carrier gas). The GC-MS system was
equipped with an inert capillary column (MEGA-5MS, 30 m length, 0.25 mm inner diameter,
0.25 μm film thickness). The oven temperature of the GC column remained at 32°C for
approximately 5 min, increasing to 320 °C at 5 °C $min^{-1}$. MS data acquisition was conducted
in full scan mode, scanning within the *m/z* range of 35 to 300 amu. After the analysis, both the
Tenax tubes and the GC column were cleaned. Calibration of the system was performed using
standards of specific VOCs (EPA labelled) loaded in clean tubes. The species detected by TD-
GCMS for a typical experiment (DN4) are presented in Table S2.

## 2.6    Oxidative potential (OP) measurements

The water-soluble oxidative potential (WS-OP) of redox-active aerosol components was
measured using a DTT assay system (Fig. S1) at FORTH/ICE-HT in Patras, Greece, which is
based on the semi-automated method of Fang et al. (2015). A detailed description of the system
components, operation, measurement protocol, and data treatment, is provided in the
Supplementary Information Section S1. Briefly, the fresh and aged aerosol samples (1.5 $cm^2$
punches of the collected quartz filters) are extracted, filtered, and incubated with DTT, in
excess, under controlled conditions. The DTT is gradually oxidized by ROS in the sample, with
its consumption rate (DTT activity, in nmol $min^{-1}$) determined spectrophotometrically by
measuring the absorbance of 2-nitrobenzoic acid (TNB), the derivatization product of DTT
with DTNB reagent, at 412 nm at specific time intervals. The WS-OP was calculated by
correcting for blank samples and was normalized to the OC mass of the sample, yielding net
DTT consumption rates (mass-normalized DTT activity – $DTT_m$) in pmol $min^{-1}μg^{-1}$ (Table S3).




OC was quantified via thermal-optical analysis (NIOSH-870 protocol), with an estimated
relative standard deviation of 15 ± 5% for replicate measurements.

## 3    Results and Discussion

### 3.1    Characterization of fresh olive wood emissions

Flaming conditions predominated in all experiments, as indicated by the estimated modified
combustion efficiency (MCE) that ranged from 0.91 to 0.99 (Table 1) (Li et al., 2015; Briggs
et al., 2016). The initial $PM_1$ concentration of the fresh olive wood burning emissions in the
chamber varied from 47 to 177 μg m$^{-3}$ (considering experiments DN1-DN8 and ND1-ND6).
This range of concentrations is representative of light to severe biomass burning pollution
episodes in polluted urban areas during wintertime (Chen et al., 2022; Luo et al., 2022; Othman
et al., 2022). The average AMS collection efficiency (CE) of the fresh emissions averaged 0.8
± 0.2, while the mean OA density, calculated following the approach of Kostenidou et al.
(2007), was 1.11 ± 0.12 g cm$^{-3}$. Estimating the OA density from measured O:C and hydrogen-
to-carbon (H:C) ratios, following the Kuwata et al. (2012) approach, yielded an average of 1.18
± 0.03 g cm$^{-3}$.

The fresh aerosol primarily consisted of organics (95 ± 3 %) with OA concentrations
ranging from 46 up to 174 μg m$^{-3}$ (Table 1). The rest of the aerosol consisted of BC (2.4 ±
2.4%), nitrates (1.4 ± 0.7%), sulfates (0.7 ± 0.4%), chloride (0.4 ± 0.2%) and ammonium (0.2
± 0.1%). In experiment DN3, ammonium sulfate seeds were also present explaining the higher
initial sulfate (28%) and ammonium (10%) content.

The initial mass ratio of the organic aerosol to black carbon (OA/BC) ranged from 13 to
263. The OA/BC differs significantly depending on the combustion conditions. When MCE
values exceed 0.9, the OA/BC ratio can range between 0.3 to 10$^5$ (McClure et al., 2020), with
higher values indicating more efficient combustion (Novakov et al., 2005). Our OA/BC values
indicate relatively efficient wood stove operation.

The average initial oxygen to carbon ratio (O:C) of the bbOA in all olive wood burning
experiments was 0.39 ± 0.04. The average initial hydrogen to carbon ratio (H:C) was 1.67 ±
0.04 ranging from 1.62 to 1.76. These values are consistent with previously reported field and
smog chamber O:C and H:C observations for fresh biomass burning aerosols (Ng et al., 2010;
Sun et al., 2016; Lim et al., 2019; Kodros et al., 2020; He et al., 2024). The relatively low AMS
$f_{44}/f_{60}$ ratios (1.56 ± 0.52) observed in the experiments are representative of fresh biomass



burning emissions from wildfires and laboratory wood burning chamber studies (Li et al., 2023).

The average high-resolution (HR) fresh bbOA mass spectrum obtained by AMS for the olive wood burning experiments (Fig. S2a) showed predominant fragments at $m/z$ 29 ($CHO^+$, $C_2H_5^+$), 41 ($C_2HO^+$, $C_2H_3N^+$, $C_3H_5^+$), 43 ($C_2H_3O^+$, $C_3H_7^+$), 55 ($C_3H_3O^+$, $C_4H_7^+$), 57 ($C_3H_5O^+$), 69 ($C_5H_9^+$, $C_4H_5O^+$) and 73 ($C_3H_5O_2^+$), suggesting a significant presence of alkenes, alkanes, and fatty acids. The observed signals at $m/z$ 44 ($CO_2^+$) and $m/z$ 60 ($C_2H_4O_2^+$), are typical tracer fragments for OOA and bbOA, respectively. The obtained fresh bbOA spectrum profile is quite similar to those reported in previous biomass burning chamber studies that examined wood or pellets burning (He et al., 2010; Kodros et al., 2020, 2022; Florou et al., 2023). The average theta angle $\theta$ of the fresh bbOA spectra, calculated for all possible pairs of the olive wood burning experiments in the present study, was on average 9° ± 7° (Fig. S3), indicating a generally similar composition of fresh bbOA.

Based on PTR-MS measurements, oxygen-containing compounds contributed the largest portion of the protonated VOCs identified in the fresh emissions (Fig. 2a). Aldehydes, including acetaldehyde ($m/z$ 45; 12.9 ± 3.7 ppb), formaldehyde ($m/z$ 31; 1.6 ± 0.7 ppb), acrolein ($m/z$ 57; 3.5 ± 1.5 ppb), and hexenal ($m/z$ 99; 2.1 ± 1.5 ppb), along with saturated ketones like acetone ($m/z$ 59; 4.7 ± 2.0 ppb) and unsaturated ones such as ethyl vinyl ketone ($m/z$ 85; 2.1 ± 1.4 ppb), contributed a total of 32.5 ppb, accounting for 19.7% of the measured VOCs. Carboxylic acids, such as formic ($m/z$ 47) and acetic ($m/z$ 61) acids, averaged a total concentration of 8.2 ppb, comprising 5% of the total VOCs (Fig. 2a). The main identified alcohol was 1-butanol ($m/z$ 75), which accounted for 3% of the VOC composition, with concentrations varying from 2.1 ppb to 9.5 ppb across experiments (Table S1). Furans and their derivatives (m/z 69, 83, 113, 147) had an average concentration of 7.9 ppb, accounting for 5% of the total measured VOCs (Fig. 2a).

Cyclic and heterocyclic aromatic compounds (with 1-ring or 2-ring structure) contributed approximately 10% to the total VOCs. This includes benzene ($m/z$ 79; 1.5 ± 1.0 ppb) and its substituted forms ($m/z$ 139, 151, 155; 3.3 ppb), toluene ($m/z$ 93; 1.2 ± 0.8 ppb), phenol ($m/z$ 95; 2.1 ± 1.7 ppb) and its substituted forms ($m/z$ 121, 135, 149, 169; 2.4 ppb in total), and C8 aromatics, including xylenes, ($m/z$ 107; 3.2 ± 2.4 ppb). Other minor contributors, with varying concentrations across experiments, included terpenes and terpenoids ($m/z$ 81 and $m/z$ 137), averaging 2.9 ppb, and naphthalene ($m/z$ 129), averaging 1.4 ± 1.1 ppb. The presence of these aromatic species is corroborated by the Tenax samples, along with compounds like benzonitrile, trimethoxy- benzene, methylindene and benzofurans. For a typical sample of fresh



emissions, chromatographic analysis yielded a variety of phenolic species other than phenol, with functional groups including several alkyl groups (methyl-, dimethyl-,ethyl), but also with oxygenated functional groups (methoxy-, dimethoxy-) as presented in Table S2. Furans comprised approximately 11% of the identifiable VOCs in the offline analysis, with the most prominent being furfural, followed by methyl-furans and methyl- furancarboxaldehyde. In terms of polyaromatic species, similarly to the PTR-QMS observations, the most abundant was naphthalene, while there were several alkyl-substituted naphthalenes present in comparable concentrations. Trace amounts of higher ring number PAHs (e.g., phenanthrene) were also observed. Most of these compounds have been previously reported in biomass burning ambient and laboratory studies (Stockwell et al., 2014; Bruns et al., 2017; Sun et al., 2019; Desservettaz et al., 2023; Florou et al., 2023).

The average WS-OP of the fresh olive wood burning aerosol was $42.9 \pm 16.1$ pmol $min^{-1}$ $\mu g^{-1}$, comparable to toxicity levels reported in literature for the water- and methanol-soluble portion of freshly emitted bbOA, which were also estimated using the acellular DTT assay protocol (Cao et al., 2021; Wang et al., 2023). The WS-OP values ranged from $21.2 \pm 5.7$ pmol $min^{-1}$ $\mu g^{-1}$ (in DN3) to $79 \pm 11.3$ pmol $min^{-1}$ $\mu g^{-1}$ (in DN7) (Table S3).

### 3.2    Effect of pine kindling on fresh olive wood emissions

In experiments ND7 and ND8, where pine kindling sticks were mixed with olive wood logs, the $PM_1$ concentration during the characterization period was 126 $\mu g$ $m^{-3}$ and 276 $\mu g$ $m^{-3}$, respectively (Table 1). High amounts of BC (67 $\mu g$ $m^{-3}$ and 190 $\mu g$ $m^{-3}$) were produced in these experiments, constituting more than half (53% and 69%) of the total fresh $PM_1$ mass. Given the efficient combustion conditions (MCE ranged from 0.96 to 0.98), these elevated BC levels were likely related to the properties of the pine (e.g., higher moisture, ash, and carbon content) compared to the olive logs (Nyström et al., 2017; Trubetskaya et al., 2021). The initial O:C of the fresh bbOA was 0.23 in experiment ND7 and 0.36 in ND8. The O:C in ND7 was the lowest of all experiments.

Comparison of the average fresh bbOA mass spectrum from olive-pine mixed emissions with that of olive logs burning (Fig. S4a) reveals significantly higher peaks at $m/z$ 28 ($CO^+$; +69%), 41 (+36%), 44 (+40%), and 73 (+39%), indicating an increase in certain oxygenated organic species. Additionally, the stronger fractional signals at $m/z$ 91 ($C_7H_7^+$; 104%), and at higher masses, such as $m/z$ 105 ($C_8H_9^+$; 154%), 129 ($C_{10}H_9^+$; +166%), suggest a higher relative contribution of cyclic hydrocarbons, PAHs, and other aromatic compounds. The theta angle of the two average fresh spectra was approximately 20°, implying distinct chemical composition



of olive-pine mixed emissions. For the VOCs, while most aromatic compound concentrations
were lower in the mixed fuel emissions, their relative contribution to the total VOCs was higher
(17.7% vs. 9.6% in olive wood alone), suggesting differences in pyrolysis pathways and
thermal degradation mechanisms between the two wood types (Fig. 2a,b). Additionally,
monoterpenes ($m/z$ 137 and their fragment $m/z$ 81) showed a significant increase in the mixed
emissions, rising from 2.9 ppb to 9.1 ppb, highlighting the influence of pine higher terpene
content on VOC composition (Fig. 2). The variations observed in aldehydes, ketones, and
heavier PAHs were within the experimental uncertainty. A more detailed breakdown of the
absolute and $CO_2$-normalized VOC concentrations, including experiment-specific observations
and comparisons, is provided in the Supplement (Fig. S5, Table S1).
No changes were observed in the WS-OP of the fresh olive-pine mixed emissions
compared to fresh olive wood emissions. The corresponding $DTT_m$ values in experiments ND7
and ND8 were $44.7 \pm 4.0$ pmol min$^{-1}$ µg$^{-1}$ and $41.1 \pm 3.4$ pmol min$^{-1}$ µg$^{-1}$, respectively (Table
S3). Similar WS-OP values (25 to 45 pmol min$^{-1}$ µg$^{-1}$) were reported by Wang et al. (2023) for
fresh bbOA from pine combustion under smoldering conditions (MCE=0.61). These values are
comparable to the average WS-OP measured in this study for olive wood emissions ($42.9 \pm$
16.1 pmol min$^{-1}$ µg$^{-1}$).

### 394 3.3 Typical day-to-night (DN) aging experiment

During a typical dry daytime-first oxidation experiment (DN1), two hours before the start of
oxidation (at t = -2 h), $70 \pm 0.4$ µg m$^{-3}$ of fresh olive wood burning $PM_1$ (91 % OA) were
injected into the chamber along with approximately 14 ppb of $O_3$ (Fig. 3). During the emissions
equilibration period (-2 to 0 h), the average O:C was 0.43, H:C was 1.67, OA/BC was 17, and
the $f_{44}/f_{60}$ ratio was 1.37 (Table 1), and remained quite stable. The WS-OP of the fresh aerosol
was estimated at $51.4 \pm 4.7$ pmol min$^{-1}$ µg$^{-1}$ (Table S3).
At time zero (t = 0 h), daytime oxidation of the emissions was initiated by turning on the
UV lights of the chamber, without adding any oxidants, and allowing the process to proceed
for 2 h. Under the given experimental conditions, each hour of UV exposure in the simulation
chamber corresponds to approximately 2 hours of atmospheric photochemical oxidation,
assuming an average OH concentration of $1.5 \times 10^6$ molecule cm$^{-3}$ (Liu et al., 2018; Nault et al.,
2018). In DN1, the average OH concentration during this 2-h oxidation period, estimated from
the decay of d9-butanol, was $3.2 \times 10^6$ molecules cm$^{-3}$, corresponding to an equivalent daytime
exposure of 4.3 h. The average $O_3$ concentration was $33 \pm 14$ ppb.



During this 2-h period the OA (wall loss corrected) increased by 22 μg m$^{-3}$ (34%). Organic nitrates also increased by 54% and $O_3$ reached 56 ppb. The H:C decreased by 4% while the $f_{44}/f_{60}$ more than doubled (3 times higher). The O:C increased from 0.43 to 0.58 (35%), consistent with previous observations (Tiitta et al., 2016). The change in the HR-AMS spectrum of the day-aged OA was modest ($\theta = 8°$). The photochemical processing resulted in an 50% increase of WS-OP (77.6 ± 6.3 pmol min$^{-1}$ μg$^{-1}$) of the bbOA (Table S3). Similar increases of OP have also been reported in previous studies (Wong et al., 2019; Lei et al., 2023; Wang et al., 2023).

Furans, terpenes and cyclic aromatic hydrocarbons, major precursors of SOA production, were significantly reduced during daytime (Fig. S6). Aromatic hydrocarbons including toluene ($m/z$ 93), phenol ($m/z$ 95), styrene ($m/z$ 105), C8 aromatics ($m/z$ 107), C9 aromatics ($m/z$ 121), and creosol/2-methoxy-4-methylphenol ($m/z$ 139) reacted and their levels were reduced (Fig. S6b). Daytime aging also led to small changes (1 ppb or less) in the concentrations of formaldehyde, acetaldehyde, acetone, acetic acid, and heptanal which however could also be attributed to chamber background effects. According to TD-GCMS analysis, maleic anhydride was also identified at $m/z$ 99 in the aged emissions (Table S2).

Reaction with OH radicals was estimated to be the dominant daytime oxidation pathway for most of the examined VOC species. For methyl vinyl ketone ($m/z$ 71), benzene ($m/z$ 79), monoterpenes fragment ($m/z$ 81), methyl furan ($m/z$ 83), toluene ($m/z$ 93), phenol ($m/z$ 95), and C8 aromatics ($m/z$ 107, assuming o-xylene), the observed reductions in concentration were close to the theoretically expected values (Table S4). Lower than predicted reductions, due to OH oxidation, were observed for furan/isoprene ($m/z$ 69; 32% less), ethyl vinyl ketone ($m/z$ 85; 21% less), styrene ($m/z$ 105; ~30% less), C9 aromatics ($m/z$ 121; assuming 1,2,3 trimethylbenzene; 23% less), monoterpenes ($m/z$ 137; assuming a-pinene; 58% less), and creosol ($m/z$ 139; 60% less). This discrepancy from theoretical predictions is likely due to the presence of other compounds at the same $m/z$ signal, including isomers, that react more slowly. Ozone-induced oxidation was a minor consumption mechanism for most of the VOCs ($k_{O_3}$ ranged from $10^{-17}$ to $10^{-22}$ molecule$^{-1}$ cm$^3$ s$^{-1}$) (Table S5), with the exception of monoterpenes and their fragments ($m/z$ 137 and 81).

At the end of the daytime oxidation (t = 2 h), the UV lights were turned off, and nighttime oxidation of the already aged emissions was conducted for two hours (2–4 h) by injecting additional 80 ppb of $O_3$ and 130 ppb of $NO_2$ into the chamber. The reaction of $NO_2$ and $O_3$ resulted in the decrease of their levels along with production of $NO_3$ radical (Fig. 3d). Although

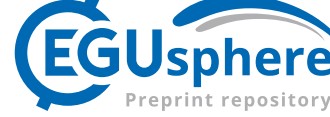

the NO$_3$ radical concentration was not directly measured in this study, it was estimated to range
between 1 and 5×10$^8$ molecule cm$^{-3}$ (typical for nighttime urban environments) based on
previous dark aging experiments conducted with the same chamber setup under similar
conditions (Kodros et al., 2022; Florou et al., 2023). This corresponds to approximately 4–7
hours of equivalent atmospheric exposure.
Nighttime aging led to further bbSOA production, with OA increasing by 17%, reaching
100 μg m$^{-3}$ (Fig. 3a). Organic nitrate increased by 0.62 μg m$^{-3}$ (72%) and total nitrate by 0.94
μg m$^{-3}$ (53%) compared to their daytime levels. Nighttime enhancement of organic nitrate has
been also reported in other studies (Kodros et al., 2020, 2022; Florou et al., 2023). The O:C
ratio sightly increased from 0.58 to 0.61 (5%). A small decrease (<1%) in H:C was observed,
while the $f_{44}/f_{60}$ increased further by 19% due to the nighttime oxidation. At the end of the
daytime-first oxidation cycle the theta angle of the aged aerosol compared to the fresh one was
23° (Fig. 3e), suggesting significant differences. The final DTT$_m$ of the aged emissions was
68.7 ± 6.0 pmol min$^{-1}$ μg$^{-1}$, higher by 12% compared to the daytime measured WS-OP and 34%
higher than the fresh one (Table S3).
The most notable VOC increases were observed for formaldehyde (m/z 31), which
increased from 1.8 to 2.2 ppb (22%); hexenal/maleic anhydride (m/z 99), which increased from
2.7 to 3.7 ppb (37%), and 2,3-benzofurandione (m/z 149), that increased from 0.2 to 0.3 ppb
(50%) (Fig. S6). Further decreases in the gas-phase concentrations of furan (m/z 69) by 0.4 ppb
(61%), methylfuran (m/z 95) by 0.35 ppb (26%), phenol (m/z 95) by 1.1 ppb (86%), and styrene
(m/z 105) by 0.4 ppb (56%), were observed (Fig. S6).

### 3.4  Typical night-to-day (ND) aging experiment

ND1 is as a typical night to day oxidation experiment (Fig. 4). The initial PM$_1$ concentration
injected into the chamber was 121 μg m$^{-3}$, with OA contributing 97%. The transition from fresh
emissions to nighttime (0-2 h) and then daytime (2-4 h) oxidation resulted in significant
changes in both the particle and gas phase. OA concentration increased by 78 μg m$^{-3}$ (65%
increase) during the nighttime oxidation and by 34 μg m$^{-3}$ (an additional 17% increase) during
daytime oxidation. During nighttime, total nitrate increased from 0.66 to 5.4 μg m$^{-3}$, driven by
production of organic nitrate. During daytime, organic nitrate levels decreased slightly (8%)
compared to nighttime. The nighttime-first cycle also led to increases in ammonium levels first
by 0.9 μg m$^{-3}$ (from 0.3 to 1.2 μg m$^{-3}$) and then by 0.3 μg m$^{-3}$ (from 1.2 to 1.5 μg m$^{-3}$).
The theta angle between the HR-AMS spectra of fresh and night-aged OA was 13° while
at the end of the nighttime-first oxidation cycle the overall change of spectrum of the aged



aerosol compared to the fresh one was 24° (Fig. 4e), similar to that observed during the daytime-first cycle. The $f_{44}/f_{60}$ ratio increased from 1.7 to 5.5 during night and from 5.5 to 9 during the day, while the H:C decreased from 1.67 to 1.61 and 1.58, respectively (Tables 1 and 2). The O:C increased by 34% (from 0.41 to 0.55) at night with a further 20% enhancement (from 0.55 to 0.66) observed after the day aging. $DTT_m$ increased from $31.8 \pm 2.8$ pmol min$^{-1}$ µg$^{-1}$ for the fresh aerosol to $42.5 \pm 3.1$ pmol min$^{-1}$ µg$^{-1}$ for night-aged aerosol (36% increase) and to $71.0 \pm 5.7$ pmol min$^{-1}$ µg$^{-1}$ for day-aged aerosol (67% increase) (Table S3). Unlike experiment DN1, which exhibited an initial increase (daytime) followed by a decrease (nighttime) in oxidative potential, experiment ND1 showed a monotonic increase with aging (Fig. 5).

During daytime oxidation, the OH concentration was $4.2 \times 10^6$ molecule cm$^{-3}$, which corresponds to approximately 5.3 h of equivalent photochemical atmospheric aging. O$_3$ levels increased by 88 ppb, rising from 96 ppb to 184 ppb by the end of daytime oxidation.

Similar trends and concentration levels were observed for most identified VOCs in experiments ND1 and DN1 (Fig. S6a and Fig. S7a). The observed differences in the percentage reduction of key bbSOA precursors, such as aromatic compounds and furans, between DN1 and ND1 (Fig. S6b and Fig. S7b) suggest that the variability in precursor depletion dynamics is primarily influenced by differences in oxidant availability, photochemical reactivity, and the chemical composition of the emissions.

### 3.5 Results of all dry daytime-first and nighttime first experiments

The average OA production (including organic nitrate) observed in the daytime-first (DN1-DN8) and nighttime-first (ND1-ND6) experiments at the end of a complete diurnal aging cycle was $51 \pm 22$ µg m$^{-3}$, ranging from 19 to 136 µg m$^{-3}$ (Fig. 6). These values correspond to a total OA mass increase ranging from 35% to 91% compared to the fresh OA. In both oxidation cycles the majority of the produced OA was formed during the first stage of oxidation. This is consistent with the higher availability of precursor VOCs initially. No significant differences were observed in the levels of OA produced in experiments ND7 and ND8 compared to the rest (Fig. 6). This suggests that, although the use of pine kindling resulted in a different composition of fresh wood emissions, its overall impact on SOA production was less significant compared to the influence of oxidation conditions.

In some cases, the nighttime-first oxidation cycle resulted in higher SOA production (Fig. 6 and Fig. 7b). For instance, in experiment ND1, an OA mass increase of over 90% was observed at the end of the nighttime-first cycle. Similarly, in ND5, the OA increased by 78%



compared to fresh. Both experiments had high initial OA concentration in the fresh emissions
(Table 1) and higher initial $O_3$ levels (at around 30 ppb) compared to the rest of the nighttime-
first experiments, which had an average level of $14 \pm 2$ ppb. However, statistical analysis did
not confirm that the nighttime-first oxidation cycle generally leads to higher SOA production
compared to the daytime-first cycle.

During both daytime-first and nighttime-first oxidation cycles, the average density of the

aged aerosol increased from 1.17 to 1.33 g cm$^{-3}$, corresponding to approximately 13% increase
in both cases (Table 2). Similar increases in bbSOA density, in the range of 1.31-1.34 g cm$^{-3}$,
have been also reported in other chamber studies during dark aging (Li et al., 2015; Florou et
al., 2023).

At the end of the daytime-first oxidation cycle, the average O:C was $0.61 \pm 0.04$, 56%

higher than the average O:C ($0.39 \pm 0.03$) of the fresh bbOA in our experiments. Almost 90%
of this increase occured during daytime (O:C increased from 0.39 to 0.59; $\Delta$O:C= 0.2) (Fig.
7c), while the subsequent nighttime oxidation resulted in an additional 10% increase in O:C
(from 0.59 to 0.61). For the nighttime-first cycle, the O:C increased from $0.40 \pm 0.06$ for the
fresh emissions to $0.61 \pm 0.06$ (a 54% increase) at the end of the cycle (Fig. 7d). In this case,
the contributions of the nighttime and daytime oxidation stages to the increase in O:C were
almost equal, at 55% and 45%, respectively. In both oxidation cycles the final O:C is similar,
but the importance of each oxidation stage depends on the order (oxidation sequence).

In all experiments, the OA AMS spectra changed progressively with aging. The

predominant differences between the average fresh and aged bbSOA spectra at the end of
daytime-first cycle were found for *m/z* 28 (more than 2-fold increase) and 44 (1.5-fold increase)
(Fig. S2). Significant decreases were observed for *m/z* 60 (37%) and 91 (36%), 115 (38%) and
137 (42%). The same changes were observed comparing the fresh and the nighttime-first aged
bbSOA (Fig. S2). During the daytime-first cycles the main changes in the OA spectrum
occurred during the first (daytime) oxidation regime, with a theta angle of $26 \pm 4°$ on average
(Fig. 7e). The further change in the second step (nighttime) was $4 \pm 2°$ on average. In contrast,
a more balanced change was observed in the evolution of the theta angle over time during the
nighttime-first cycle (Fig. 7f). The average OA spectrum shifted by $19 \pm 4°$ on average during
nighttime, followed by an additional 10° shift during UV exposure. Overall, at the end of both
cycles, regardless of the followed oxidation path, the final average bbSOA spectra were almost
identical ($\theta < 3°$) (Fig. 8).

To evaluate the environmental relevance of the chamber-produced bbSOA, the final

daytime-first and nighttime-first bbSOA spectra from this study were compared to the spectra





of oxidized OA, that was measured at a remote site in Greece (Pertouli) during the summer of 2022 (Vasilakopoulou et al., 2023). Most of this aged OA was aged emissions of wildfires from different regions of Europe. Two oxygenated OA (OOA) factors; a more-oxidized OOA (MO-OOA) and a less-oxidized OOA (LO-OOA) were needed to reproduce the observed OA spectra. Our final bbSOA spectra showed greater similarity to the LO-OOA factor, with a theta angle of approximately 16°, and were more distinct (θ at around 30°) from the MO-OOA spectra measured in Pertouli. This suggests that our experiments simulated the earlier stages of atmospheric aging, while additional aging processes likely occur under ambient conditions (see also Fig. S8).

Changes in VOC levels of aged emissions across all daytime first (Fig. S9) and nighttime-first (Fig. S10) experiments were consistent with those observed in the typical experiments DN1 and ND1. Both aging cycles resulted in a significant decrease in the concentration of furans and their derivatives, cyclic and polycyclic aromatic hydrocarbons and terpenes. The day aged Tenax samples indicated a moderate decrease in aromatic species like toluene (~20%) and benzene, which is consistent with their lower reaction rates compared to higher carbon number aromatics. Rapid decrease in concentration was noted for species like phenol (~45%) and furfural (~75%), as well as their structurally related compounds. Related products, including 2-nitro-phenol, 4-methyl-2-nitro-phenol, maleic anhydride, and 3-methyl-2,5-furandione, were also detected. p-Benzoquinone was also formed, possibly as a result of the reacted aromatics. Benzofuran was absent from the aged samples; instead, 2,3-benzofurandione was detected. At the same time a progressive increase in aldehydes and ketones was observed, along with significant increases in carboxylic acids, such as formic (*m/z* 47) and acetic (*m/z* 61). The benzaldehyde concentration increased, accompanied by the formation of benzeneacetaldehyde, 2-hydroxy-benzaldehyde, 3-ethyl-benzaldehyde. A notable increase in butanol was also observed in the Tenax samples, along with the formation of straight-chain aldehydes (hexanal to undecanal). The GC-MS measurements for the night-aged samples following daytime processing were consistent with those of the PTR-MS. Furfural was no longer detected, while a further decrease in phenol and increases in benzaldehyde and butanol were noted. A cumulative depiction of the experiment's progression in terms of oxidation and VOCs detected by the GC-MS, is provided in Fig. S11. Similar results were obtained for the other experiments.



### 3.6 Effect of daytime-first and nighttime-first oxidation cycle on WS-OP

The water-soluble oxidative potential (WS-OP) of fresh emissions ranged from 21 pmol min$^{-1}$ µg$^{-1}$ to 79 pmol min$^{-1}$ µg$^{-1}$ and that of aged wood-burning emissions from 39 pmol min$^{-1}$ µg$^{-1}$ to 127 pmol min$^{-1}$ µg$^{-1}$ (Table S3). These values fall within the range reported in literature for fresh bbOA and aged bbOA (Verma et al., 2015; Tuet et al., 2017; Bates et al., 2019; Daellenbach et al., 2020; Wang et al., 2023).

The evolution of average WS-OP of fresh and aged emissions, considering all experiments and both oxidation cycles, (Fig. 9) was similar to that observed in experiments DN1 and DN1 (Fig. 5). The average WS-OP values for the daytime-first cycle were 47.9 ± 17.7 pmol min$^{-1}$ µg$^{-1}$ for fresh emissions, 93 ± 27 pmol min$^{-1}$ µg$^{-1}$ for daytime-aged emissions, representing a 94% increase compared to fresh aerosol, and 73.4 ± 13.3 pmol min$^{-1}$ µg$^{-1}$ for nighttime-aged emissions, indicating a 21% reduction compared to daytime-aged WS-OP (Fig. 9a). For the nighttime-first oxidation cycle, the average WS-OP of the fresh emissions were 37.8 ± 10.6 pmol min$^{-1}$ µg$^{-1}$. After nighttime aging, it increased by 44% to 54.4 ± 13.6 pmol min$^{-1}$ µg$^{-1}$, and following daytime aging, it further increased by 62.9 ± 20.4 pmol min$^{-1}$ µg$^{-1}$ (Fig. 9b).

Statistical analysis (t-test) showed that aged WS-OP values were significantly higher than those of fresh emissions in all experiments, for both oxidation cycles. Additionally, for the daytime-first oxidation cycle, a statistically significant difference was observed between the WS-OP of nighttime-aged (NO$_3$-oxidized) and daytime-aged (UV-oxidized) emissions. Further details on the statistical analysis are provided in Supplementary Section S4 and Table S9.

The overall increase in WS-OP at the end of the two oxidation cycles was 53 ± 34% for the daytime-first cycle and 66 ± 8% for the nighttime-first cycle, indicating that both daytime and nighttime aging of biomass burning emissions consistently enhanced their oxidative potential. Our results suggest that the sequence of chemical processes – whether the emissions are first oxidized by OH or NO$_3$ – can significantly affect the temporal evolution of OP. This, in turn, may also influence the health impacts associated with exposure to biomass burning plumes, depending on the time of day when the emissions occur. Although daytime boundary layer dynamics generally favour mixing and dilution of pollutants, daytime burning in urban environments may actually be as or more aggravating than nighttime burning, owing to the enhanced oxidative processing of the emissions occurring in the former stage of the diurnal cycle.



The correlation of WS-OP with produced OA and degree of oxidation (O:C) were also
investigated. Three OA types were considered (fresh, day-aged, and night-aged). WS-OP was
not well correlated with either the O:C ratio ($R^2 < 30\%$) of the organic aerosol or its fresh and
aged fractions ($R^2$ up to 34%) (Fig. S12). This implies that the link between bbOA aging, and
WS-OP change is complex and cannot be just described by one variable.
The observed WS-OP trends could be linked to the VOC composition and oxidation
processes in the daytime-first and nighttime-first cycles. The daytime-first cycle exhibits a high
daytime WS-OP due to the OH oxidation of VOCs such as furans, aromatics, and phenolic
compounds, leading to the formation of reactive species like, 4-methyl-2-nitrophenol, and
highly reactive p-benzoquinone. In contrast, the nighttime-first cycle shows a gradual increase
in WS-OP. After one complete diurnal cycle, WS-OP values in both cycles converge,
indicating that oxidative processes in both pathways ultimately lead to similar levels of
oxidation products. This convergence highlights the role of both fast and slow oxidation
mechanisms in determining aerosol OP and suggests that even VOCs with lower reactivity can
significantly contribute to aerosol toxicity over extended atmospheric aging.

## 4   Conclusions

This study investigated how different diurnal oxidation sequences, daytime-first and
nighttime-first, affect the formation of OA, the gas-phase composition, and the oxidative
potential of emissions produced by burning olive wood and olive wood mixed with pine
throughout a complete diurnal aging cycle. Both daytime-first and nighttime-first oxidation
cycles resulted in enhancement in OA levels by 35%-90%. The mixture of olive wood with
pine kindling resulted in a different composition of fresh emissions; however, its overall impact
on SOA production was less significant compared to the influence of oxidation conditions.
The daytime-first cycle favoured rapid daytime oxidation, producing highly oxygenated
species and increasing the O:C ratio of the fresh emissions from $0.39 \pm 0.04$ to $0.59 \pm 0.04$
during daytime, reaching finally at $0.61 \pm 0.03$ during nighttime. The nighttime-first cycle
showed a gradual (two-steps) oxidation increase with a similar final O:C ratio of $0.61 \pm 0.06$.
The daytime-first cycle exhibited rapid spectral changes during daytime oxidation, while
nighttime-first cycles showed a more balanced two-step evolution. At the end of both cycles,
the final average bbSOA spectra were nearly identical ($R^2 > 0.99$; $\theta < 3°$), indicating that the
aerosol was transformed into similar aged OA regardless of the initial oxidation step (daytime
or nighttime) at the start of the cycle. The chamber-produced bbSOA resembled the less-
oxidized OOA in a field campaign in Greece with the corresponding OA dominated by aged



bbOA, suggesting that the present study has addressed only part of the aging that occurs in the atmosphere.

Both the daytime-first and nighttime-first oxidation cycles effectively reduced the concentration of bbSOA precursors (e.g., furans, aromatic hydrocarbons, terpenes). Concurrently, a progressive increase in aldehydes and ketones was observed in both cycles, alongside increases in carboxylic acids, such as formic and acetic acids. The daytime-first cycle resulted in a $53 \pm 34\%$ increase in WS-OP of aerosol while the nighttime-first cycle showed a slightly higher increase of $66 \pm 8\%$. The final WS-OP values of the daytime-first ($73 \pm 14$ pmol min$^{-1}$ µg$^{-1}$) and nighttime-first ($63 \pm 20$ pmol min$^{-1}$ µg$^{-1}$) cycles were statistically similar.

These findings underscore the importance of considering oxidation sequences when assessing the environmental fate and health impacts of biomass burning emissions. This highlights the complex and dynamic nature of atmospheric aging processes. Future research should investigate the effects of prolonged atmospheric aging under more realistic conditions, such as higher initial relative humidity, multiple day-night cycles, and conduct detailed chemical analyses of particle-phase products to better understand contributions of specific chemical components to aerosol OP.

## Author contributions

M.P.G., K.F., and A.M. contributed to investigation, conducted the experiments and performed the laboratory measurements.; M.P.G. and G.S. performed the offline measurement of the water-soluble oxidative potential of the collected aerosol samples.; A.M. performed the offline TD-GCMS analysis of the Tenax samples; C.K. contributed to chamber set-up optimization.; A.N. conceived and supported the research project; S.N.P. supported and directed this research.; M.P.G. and K.F. interpreted the results and contributed to formal data analysis; M.P.G. wrote the original manuscript with contributions from all co-authors.; All authors contributed to the review and editing of the manuscript and have approved the final submitted version.

## Conflicts of interest

The authors declare that there are no conflicts to declare.

## Funding information

This work was supported by the project NANOSOMs (Grant 11504) of the Greek HFRI, the Horizon 2020 project REMEDIA (grant agreement no. 874753), and the European Research Council (ERC) under the European Union's Horizon 2020 research and innovation programme (grant agreement no. 726165, PyroTRACH – Pyrogenic Transformations Affecting Climate and Health).



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



**Table 1:** Initial biomass burning aerosol composition and initial chamber conditions for all the
conducted experiments.

| Exp. | PM$_1$ | BC | Ammoniun | Sulfate | Organics | Nitrate | Org. Nitrate | Inorg. Nitrate | Chloride | $f_{44}/f_{60}$ | O:C | H:C | RH$_{init}$ | MCE* |
|---|---|---|---|---|---|---|---|---|---|---|---|---|---|---|
| | [µg m$^{-3}$] | [µg m$^{-3}$] | [µg m$^{-3}$] | [µg m$^{-3}$] | [µg m$^{-3}$] | [µg m$^{-3}$] | [µg m$^{-3}$] | [µg m$^{-3}$] | [µg m$^{-3}$] | | | | [%] | |
| DN1 | 70 | 3.7 | 0.06 | 0.62 | 63.6 | 1.18 | 0.56 | 0.62 | 0.45 | 1.37 | 0.43 | 1.67 | 13 | 0.96 |
| DN2 | 114 | 7.9 | 0.13 | 0.86 | 102 | 2.01 | 1 | 1.01 | 0.42 | 2.21 | 0.39 | 1.66 | N/A | 0.92 |
| DN3 | 79.1 | N/A | 8.1 | 22.1 | 48.2 | 0.52 | 0.23 | 0.29 | 0.11 | 1.82 | 0.38 | 1.62 | N/A | 0.99 |
| DN4 | 177 | 1.1 | 0.24 | 1.11 | 172 | 2.02 | 0.91 | 1.11 | 0.62 | 1.11 | 0.39 | 1.71 | N/A | 0.91 |
| DN5 | 102 | 0.4 | 0.09 | 0.47 | 99.8 | 0.65 | 0.26 | 0.39 | 0.19 | 1.00 | 0.35 | 1.72 | N/A | 0.99 |
| DN6 | 53.6 | 0.9 | 0.07 | 1.02 | 49.6 | 1.66 | 1.02 | 0.64 | 0.41 | 1.50 | 0.36 | 1.76 | N/A | 0.96 |
| DN7 | 74.8 | 0.8 | 0.06 | 0.54 | 72.1 | 1.15 | 0.43 | 0.72 | 0.16 | 1.34 | 0.41 | 1.67 | N/A | 0.91 |
| DN8 | 85.5 | 5.4 | 0.09 | 0.49 | 78.0 | 1.22 | 0.28 | 0.94 | 0.33 | 1.02 | 0.44 | 1.65 | 13 | 0.92 |
| ND1 | 121 | N/A | 0.28 | 1.01 | 118 | 1.21 | 0.66 | 0.55 | 0.58 | 1.72 | 0.41 | 1.65 | 13 | 0.92 |
| ND2 | 72.0 | 0.5 | 0.21 | 0.45 | 69.4 | 1.13 | 0.76 | 0.37 | 0.24 | 2.53 | 0.47 | 1.67 | N/A | 0.96 |
| ND3 | 47.2 | 0.4 | 0.16 | 0.12 | 45.8 | 0.67 | 0.34 | 0.33 | 0.05 | 2.37 | 0.29 | 1.67 | 14 | 0.94 |
| ND4 | 93.3 | 1 | 0.22 | 0.54 | 90.2 | 0.99 | 0.50 | 0.49 | 0.34 | 1.61 | 0.40 | 1.66 | 15 | 0.92 |
| ND5 | 176 | N/A | 0.25 | 0.34 | 174 | 1.35 | 0.62 | 0.73 | 0.18 | 1.27 | 0.37 | 1.66 | 13 | 0.91 |
| ND6 | 124 | 3 | 0.18 | 0.41 | 120 | 0.72 | 0.31 | 0.41 | 0.38 | 0.92 | 0.43 | 1.65 | 14 | 0.90 |
| ND7 | 126 | 67 | 0.06 | 0.07 | 58.7 | 0.15 | 0.10 | 0.05 | 0.05 | 2.47 | 0.23 | 1.61 | 12 | 0.98 |
| ND8 | 276 | 190 | 0.12 | 0.62 | 83.4 | 1.22 | 0.85 | 0.37 | 0.25 | 1.88 | 0.36 | 1.65 | 24 | 0.96 |

*Modified combustion efficiency (MCE) calculated based on equation: $([\Delta CO_2]/([\Delta CO]+[\Delta CO_2])$).




**Table 2:** Composition of aged biomass burning aerosol, averaged over the last 30 minutes of each oxidation state, for both daytime-first (DN) and nighttime-first (ND) experiments.

| Exp. | Oxid. | PM$_1$ [µg m$^{-3}$] | Ammonium [µg m$^{-3}$] | Sulfate [µg m$^{-3}$] | Organics [µg m$^{-3}$] | Nitrate [µg m$^{-3}$] | Org. Nitrate [µg m$^{-3}$] | Inorg. Nitrate [µg m$^{-3}$] | Chloride [µg m$^{-3}$] | ρ* [g cm$^{-3}$] | $f_{44}/f_{60}$ | O:C | H:C |
|---|---|---|---|---|---|---|---|---|---|---|---|---|---|
| **Day - Night** | | | | | | | | | | | | | |
| DN1 | Day | 88 | 0.1 | 0.7 | 85 | 1.76 | 0.86 | 0.90 | 0.38 | 1.34 | 4.3 | 0.58 | 1.61 |
| | Night | 104 | 0.15 | 0.83 | 100 | 2.74 | 1.48 | 1.26 | 0.4 | | 5.1 | 0.61 | 1.6 |
| DN2 | Day | 157 | 0.1 | 0.9 | 154 | 2.42 | 1.75 | 0.67 | 0.26 | 1.34 | 15.3 | 0.64 | 1.52 |
| | Night | 185 | 0.47 | 1.46 | 179 | 4.21 | 2.98 | 1.23 | 0.3 | | 16.5 | 0.67 | 1.53 |
| DN3 | Day | 102 | 10.3 | 28.9 | 62 | 0.63 | 0.36 | 0.27 | 0.13 | 1.37 | 8.5 | 0.63 | 1.55 |
| | Night | 107 | 10.8 | 30.3 | 65 | 0.94 | 0.61 | 0.33 | 0.15 | | 9.7 | 0.64 | 1.55 |
| DN4 | Day | 240 | 1.1 | 1.2 | 232 | 4.92 | 2.29 | 2.63 | 0.49 | 1.32 | 3.7 | 0.58 | 1.64 |
| | Night | 262 | 1.26 | 1.35 | 252 | 6.14 | 2.82 | 3.32 | 0.52 | | 3.9 | 0.6 | 1.64 |
| DN5 | Day | 134 | 0.1 | 0.5 | 132 | 0.97 | 0.52 | 0.45 | 0.18 | 1.29 | 3.4 | 0.52 | 1.66 |
| | Night | 156 | 0.19 | 0.59 | 153 | 2.19 | 1.51 | 0.68 | 0.2 | | 4.3 | 0.56 | 1.65 |
| DN6 | Day | 83 | 0.2 | 1.3 | 78 | 2.80 | 1.55 | 1.25 | 0.38 | 1.33 | 6.5 | 0.59 | 1.63 |
| | Night | 90 | 0.22 | 1.48 | 85 | 3.27 | 1.78 | 1.49 | 0.4 | | 7.1 | 0.61 | 1.61 |
| DN7 | Day | 105 | 0.1 | 0.6 | 103 | 1.43 | 0.69 | 0.74 | 0.15 | 1.34 | 5.4 | 0.6 | 1.61 |
| | Night | 122 | 0.17 | 0.79 | 119 | 1.94 | 1.03 | 0.91 | 0.17 | | 5.8 | 0.62 | 1.61 |
| DN8 | Day | 111 | 0.1 | 0.6 | 108 | 1.68 | 0.59 | 1.09 | 0.33 | 1.33 | 3 | 0.58 | 1.61 |
| | Night | 134 | 0.19 | 0.74 | 130 | 2.95 | 1.37 | 1.58 | 0.36 | | 3.4 | 0.6 | 1.6 |
| **Night – Day** | | | | | | | | | | | | | |
| ND1 | Night | 199 | 1.2 | 1.1 | 188 | 9.0 | 5.4 | 3.6 | 0.28 | 1.36 | 5.5 | 0.55 | 1.61 |
| | Day | 233 | 1.5 | 1.3 | 222 | 8.7 | 5.0 | 3.7 | 0.29 | | 9 | 0.66 | 1.58 |
| ND2 | Night | 110 | 0.5 | 0.5 | 102 | 7.0 | 5.5 | 1.5 | 0.15 | 1.39 | 8.1 | 0.62 | 1.6 |
| | Day | 119 | 0.6 | 0.6 | 112 | 6.3 | 4.8 | 1.5 | 0.16 | | 11.1 | 0.7 | 1.57 |
| ND3 | Night | 61 | 0.2 | 0.2 | 57 | 2.9 | 2.2 | 0.7 | 0.06 | 1.28 | 7.3 | 0.42 | 1.64 |
| | Day | 66 | 0.3 | 0.2 | 63 | 2.5 | 1.9 | 0.6 | 0.07 | | 12.2 | 0.52 | 1.61 |
| ND4 | Night | 133 | 0.4 | 0.8 | 127 | 4.3 | 3.1 | 1.2 | 0.29 | 1.31 | 3.9 | 0.49 | 1.62 |
| | Day | 148 | 0.6 | 1 | 142 | 4.2 | 2.9 | 1.3 | 0.3 | | 6.4 | 0.58 | 1.6 |
| ND5 | Night | 278 | 0.5 | 0.5 | 270 | 7.0 | 4.8 | 2.2 | 0.14 | 1.31 | 3.8 | 0.48 | 1.63 |
| | Day | 315 | 0.7 | 0.6 | 307 | 6.6 | 4.0 | 2.6 | 0.14 | | 6.5 | 0.58 | 1.61 |
| ND6 | Night | 192 | 0.7 | 0.6 | 184 | 6.8 | 4.1 | 2.7 | 0.18 | 1.33 | 2.6 | 0.51 | 1.62 |
| | Day | 203 | 0.8 | 0.7 | 195 | 6.0 | 3.4 | 2.6 | 0.19 | | 4.8 | 0.61 | 1.61 |
| ND7 | Night | 182 | 0.1 | 0.1 | 93 | 4.6 | 3.1 | 1.5 | 0.06 | 1.2 | 5 | 0.36 | 1.65 |
| | Day | 211 | 0.1 | 0.2 | 105 | 4.7 | 3.0 | 1.7 | 0.08 | | 6.5 | 0.41 | 1.62 |
| ND8 | Night | 400 | 0.3 | 1.2 | 149 | 9.3 | 5.4 | 3.9 | 0.26 | 1.29 | 4.7 | 0.48 | 1.64 |
| | Day | 451 | 0.5 | 1.7 | 163 | 9.1 | 5.2 | 3.9 | 0.29 | | 7.6 | 0.54 | 1.6 |

*Density calculated based on O:C and H:C ratios, following the approach of Kuwata et al. (2012).



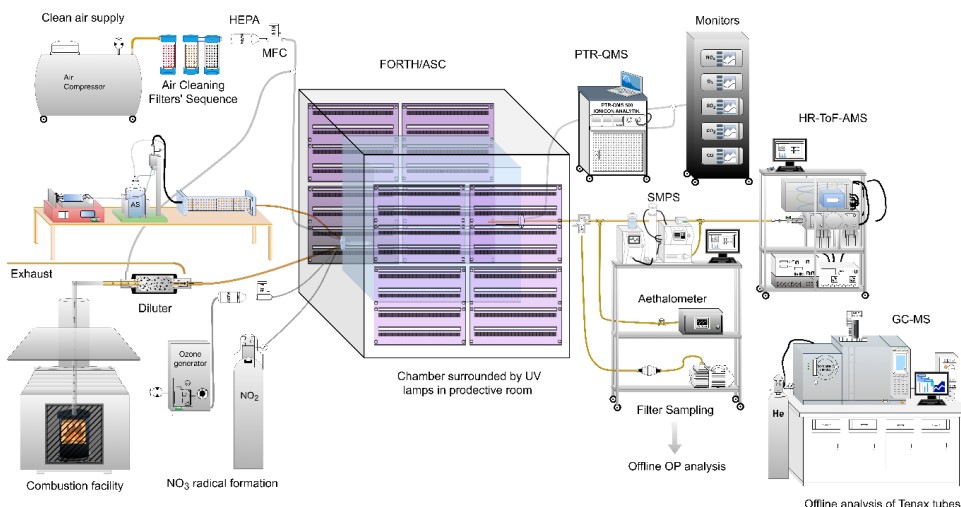


**Figure 1:** Experimental setup of the FORTH-ASC facility, illustrating the surrounding
instrumentation and the combustion facility.



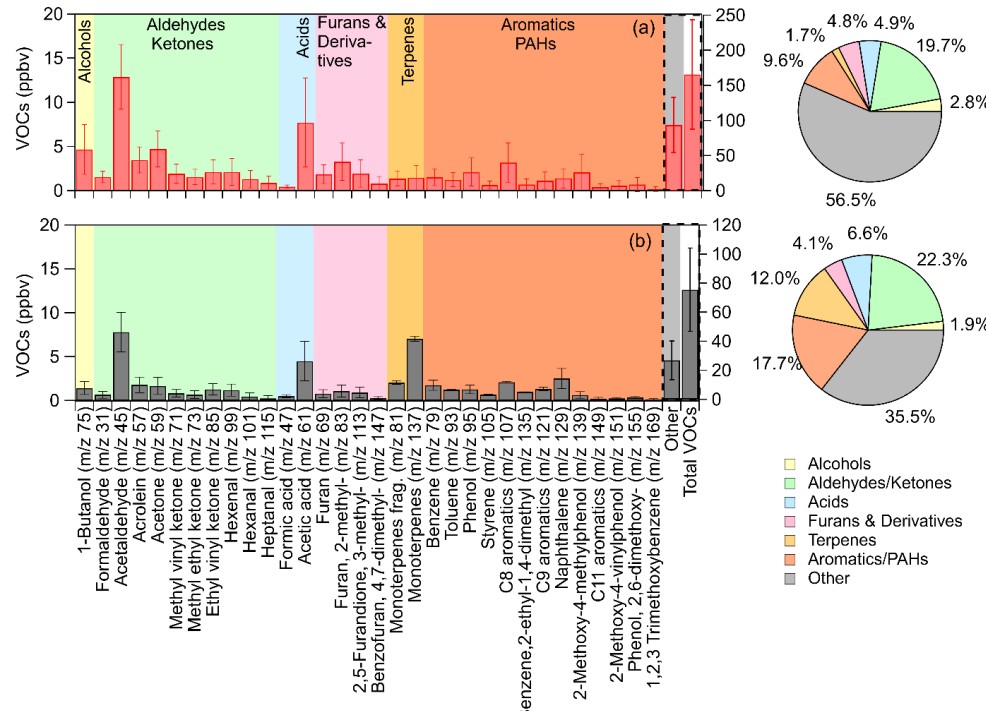


**Figure 2:** Average concentrations, in ppb, of the identified VOCs in (a) fresh olive wood burning emissions (red bars) and (b) fresh olive-pine mixed emissions (grey bars), along with their percentage contribution to the total VOCs concentration measured by PTR-QMS. The protonated *m/z* for each compound is shown in parentheses on the x-axis. The left y-axis shows the concentrations of identified VOCs, while the right y-axis displays the concentrations of the sum of the unidentified (other) and the total measured VOCs.





1080

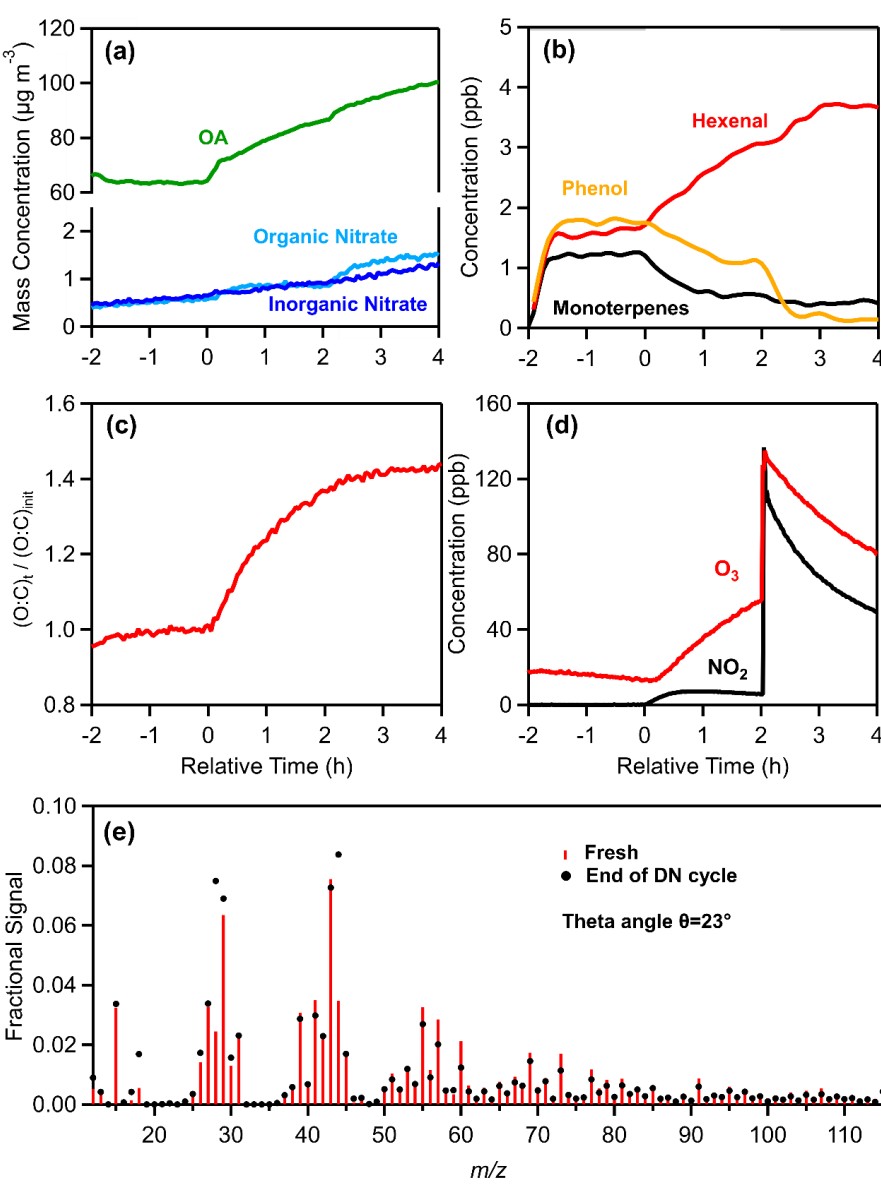

1081

**Figure 3:** Measurements from the experiment DN1, showing the time evolution of: (a) wall-loss-corrected organic aerosol, particulate organic and inorganic nitrate, (b) selected VOCs, including monoterpenes (m/z 137), hexenal (m/z 99), and phenol (m/z 95), (c) normalized O:C ratio, (d) $O_3$ and $NO_2$, and (e) a comparison of the fresh (red sticks) and nighttime (black markers) oxidized aerosol mass spectra at the end of the daytime-first oxidation cycle.





1087

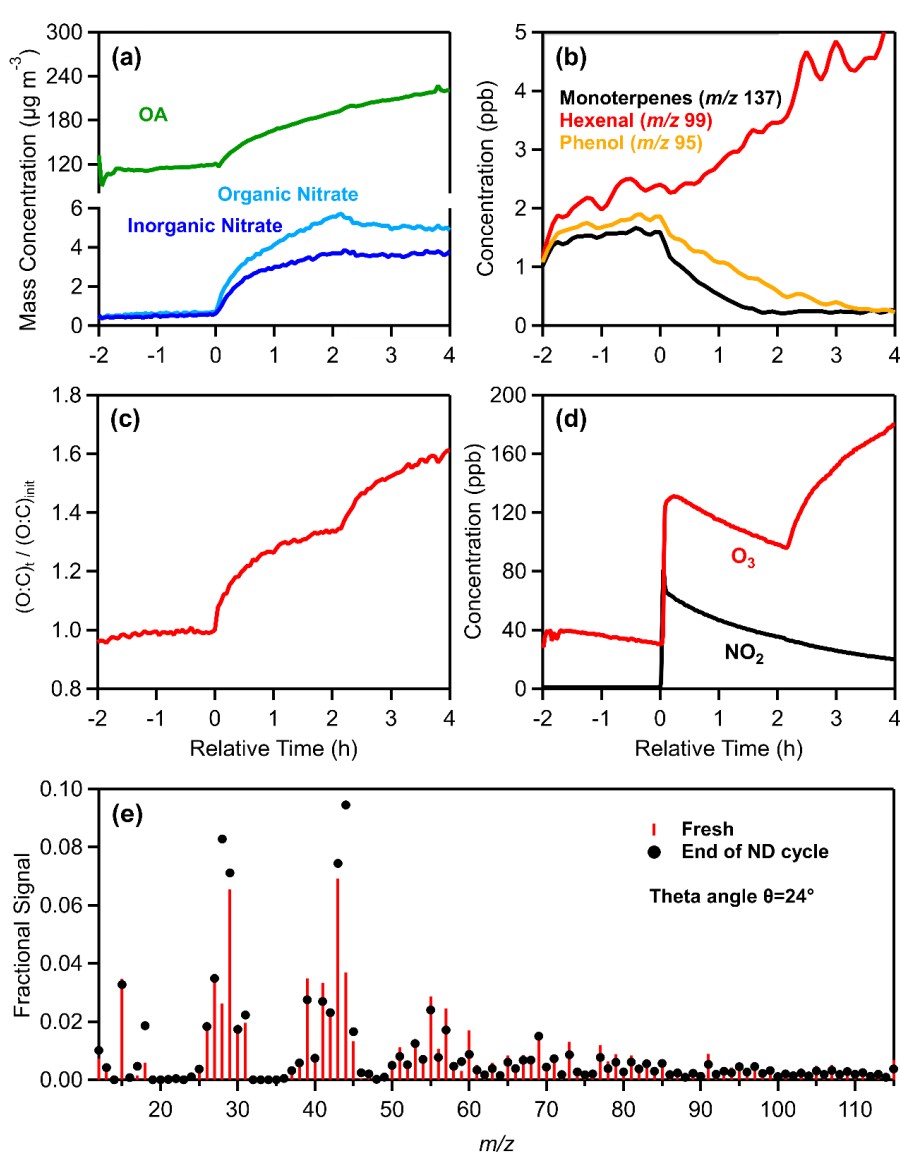

1088

**Figure 4:** Measurements from the experiment ND1, showing the time evolution of: (a) wall-loss-corrected organic aerosol, particulate organic and inorganic nitrate, (b) selected VOCs, including monoterpenes (m/z 137), hexenal (m/z 99), and phenol (m/z 95), (c) normalized O:C ratio, (d) $O_3$ and $NO_2$, and (e) a comparison of the fresh (red sticks) and daytime (black markers) oxidized aerosol mass spectra at the end of the nighttime-first oxidation cycle.






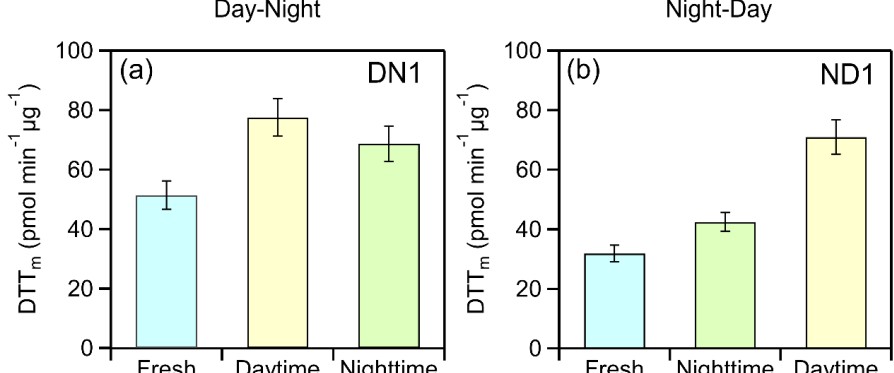


**Figure 5:** Category plots illustrating the evolution in water-soluble oxidative potential (WS-OP), expressed as per OC mass normalized $DTT_m$ activity (pmol min$^{-1}$ µg$^{-1}$), in case of typical experiment (a) DN1 and (b) ND1.


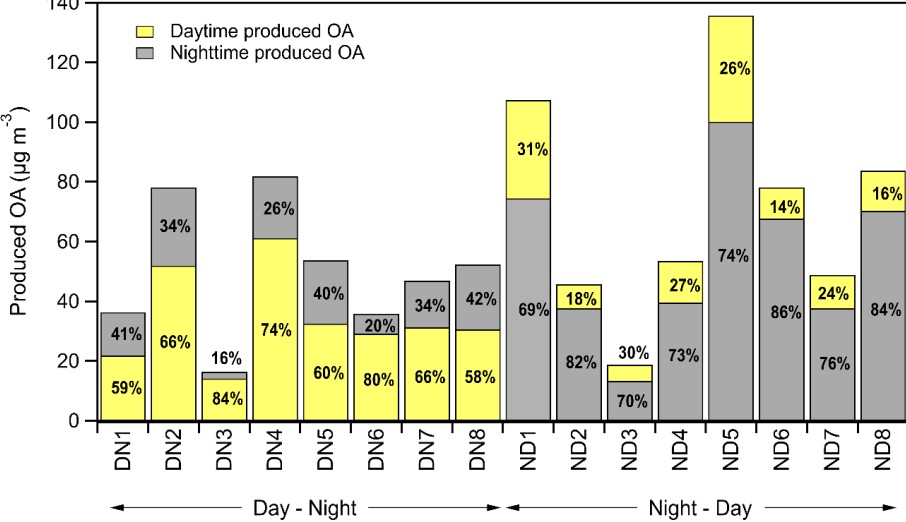


**Figure 6:** Absolute mass (in µg m$^{-3}$) and percentage increase (%) of OA (including organic nitrate) per oxidation regime (daytime, nighttime) for both daytime-first (DN) and nighttime-first (ND) cycles, for all conducted experiments.






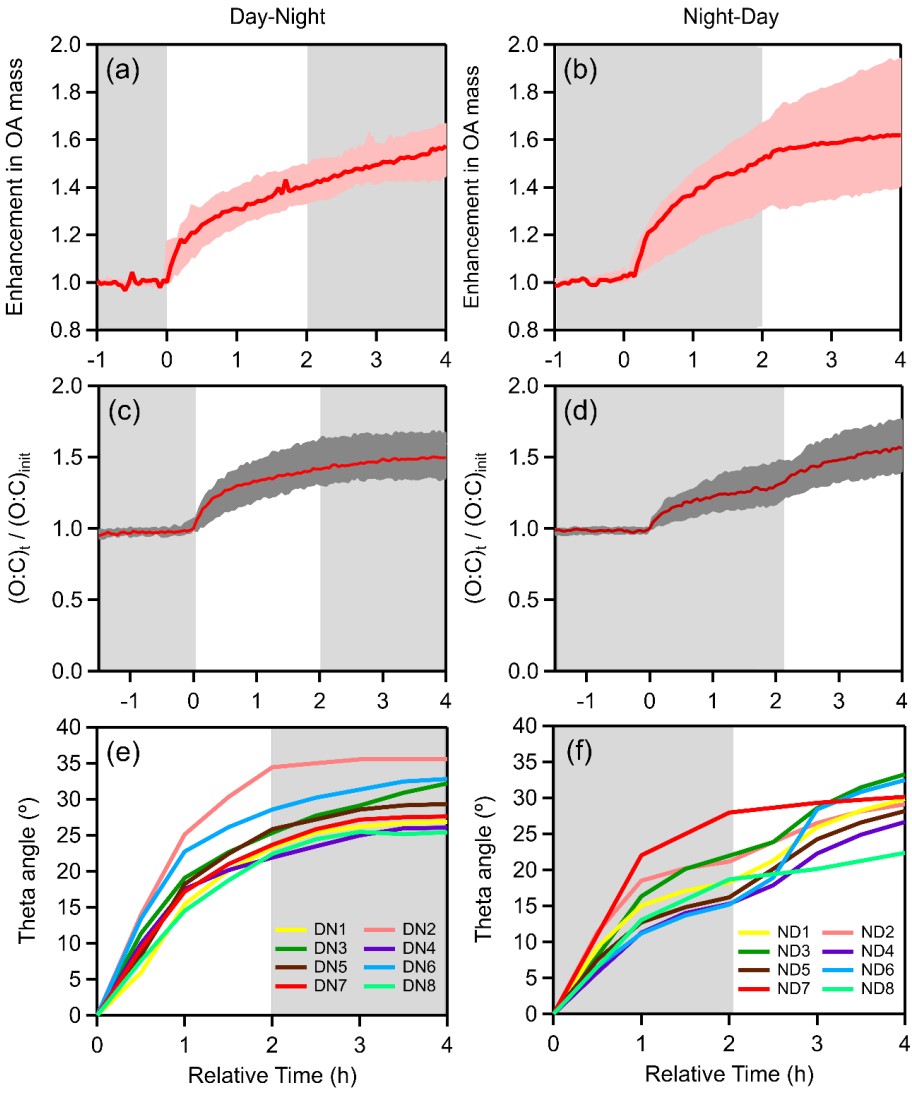


**Figure 7:** Evolution over time of: OA enhancement during (a) daytime-first and (b) nighttime-first oxidation cycle; O:C ratio enhancement during (c) daytime-first and (d) nighttime-first oxidation cycle; theta angle during (e) daytime-first and (f) nighttime-first oxidation for experiments conducted under dry initial conditions using only olive wood logs as burning fuel (DN1-DN8, ND1-ND6). In experiments ND3, ND4, and ND6, the change in spectrum occurred slightly later, as the first-step oxidation extended to 3 h compared to 2 h lasted in the other nighttime-first experiments.





1113

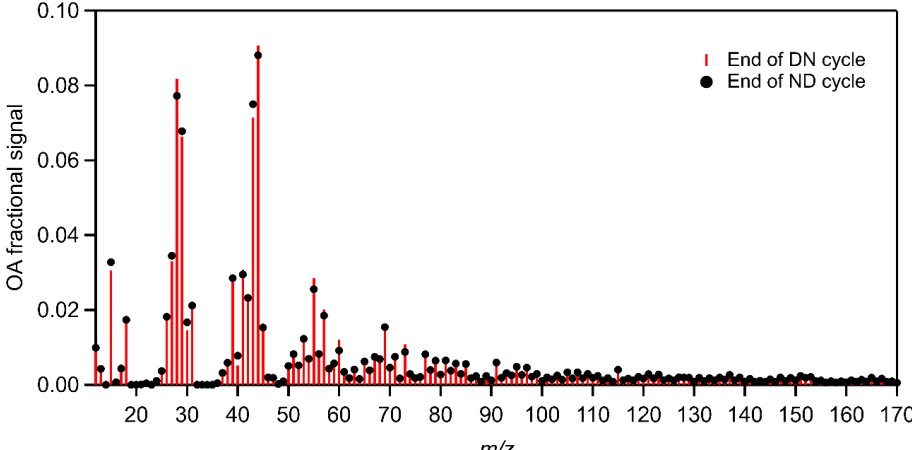

1114

**Figure 8:** Relative differences in the average spectra obtained at the end of daytime-first (DN, red sticks) and nighttime-first (ND, black circles) oxidation cycle, respectively, for experiments conducted using olive wood logs as burning fuel. The theta angle between the averaged daytime- and nighttime-first aged spectra was 3° (identical).

1119

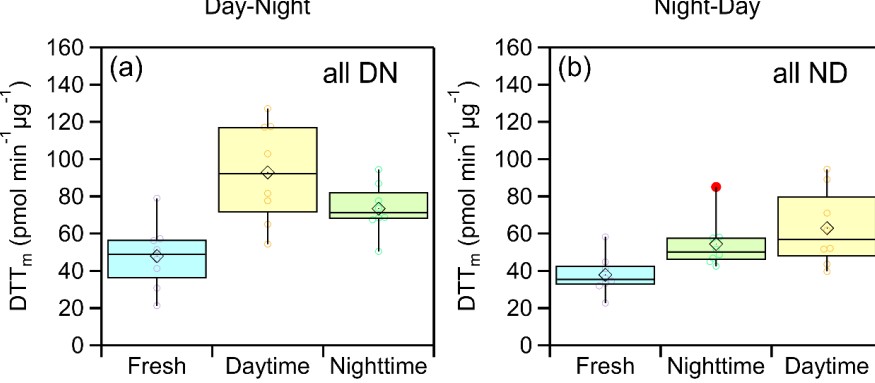

1120

**Figure 9:** Box plots illustrating the changes in WS-OP, expressed as per OC mass normalized $DTT_m$ activity (pmol min$^{-1}$ µg$^{-1}$), considering all performed experiments, in case of (a) daytime first (DN) oxidation cycle and (b) nighttime-first (ND) oxidation cycle.