# Peer review of "Diurnal aging of biomass burning emissions: Impacts on secondary organic aerosol formation and oxidative potential"

_EGUsphere, 2025_

## Author Response (AR1)

**Responses to the Comments of the Reviewers**

**Reviewer #1**

**(1)** This manuscript employed controlled atmospheric simulation chamber experiments to systematically investigate the evolution of bbOA from olive wood combustion emissions undergoing complete diurnal oxidation cycles. The core focus was the impact of the oxidation sequence. Key findings revealed that despite different initial oxidation paths, the final aged bbSOA exhibited remarkably similar chemical composition. Crucially, the study demonstrated that the temporal evolution of oxidative potential was strongly dependent on the oxidation sequence. DN cycling caused a sharp OP increase during daytime oxidation followed by a partial decrease at night, whereas ND cycling resulted in a more gradual, stepwise rise. Furthermore, WS-OP changes correlated weakly with O:C ratio, suggesting complex underlying chemical mechanisms. These findings are vital for understanding and predicting the real-world health impacts of biomass burning plumes, indicating that the timing of emission release (day vs. night) could indirectly influence ultimate toxicity by dictating subsequent chemical aging pathways. The study deserves to be published after minor revisions.

We appreciate the positive assessment of our work by the reviewer. Our responses and the corresponding changes in the manuscript (in black) follow the comments of the reviewer (in blue).

**(2)** The DTT is a key method used in this research. The authors should include a short review of how this method works in the introduction section.

Following the suggestion of the reviewer we have added a short review of how the DTT method works in the introduction section of the revised manuscript.

**(3)** Section 2.1: The authors mention many experimental procedures but lack some detailed descriptions. For example: Line 130: Why was the drying condition set to RH=12-24%, instead of RH<5%? Lines 153-158: What was the injection sequence for $NO_2$ and $O_3$? Since $NO_2$ and $O_3$ are not instantly homogeneously mixed within the smog chamber, does using $O_3$ injection as the start of the night-time experiment affect the experimental results? These details need to be described.

The low (12-24%) but not extremely low (<5%) in our experiments was due to the presence of water vapor in the wood burning emissions. Trying to remove this small amount of water, would result in losses of organic vapors and particles and would reduce the quality of our experiments without resulting in a significant change. $NO_2$ was injected first in our chamber, was allowed to mix for 10 min, and then $O_3$ was added. We have revised the Methods section to include these details.

**(4)** In the experiments, the concentrations of $NO_2$ and $O_3$ were set at levels ranging from tens to hundreds of ppb. Are these settings representative of real atmospheric conditions? How was this quantitatively assessed?

The injected concentrations of $NO_2$ are consistent with levels observed in polluted urban environments. The $O_3$ levels used are encountered in air pollution episodes during daytime and can be found in the residual layer above the nocturnal boundary layer. From there they

gradually enter the nocturnal boundary layer reacting with $NO_2$ and forming the $NO_3$ radical (Kodros et al., 2020). Using concentrations near the upper bound of real atmospheric conditions is a common practice in chamber studies, in order to accelerate the chemistry while staying under reasonable atmospheric conditions. A brief discussion of this point and the corresponding references have been added to the revised paper.

**(5)** The authors addressed particle wall loss corrections in Section 2.1. However, did they also correct for organic vapor wall losses, or were these deemed negligible? Please supplement relevant details. Additionally, was wall loss assessed for any inorganic gases, such as $NO_x$ and $O_3$?

Particle wall loss corrections were applied to all aerosol data, and organic vapor wall losses were considered negligible within the timeframe of the experiments. Wall loss of inorganic gases, such as $NO_2$ and $O_3$, was assessed during preliminary chamber characterization. These losses were found to be minimal over the course of the experiments, with typical wall losses of a few percent or less during an experiment. These issues related to wall losses have been clarified in the Methods section of the manuscript.

**(6)** Figure 6 shows higher concentrations of SOA for nighttime-prioritized experiments (e.g., ND1, ND5), but the reason for this is not explained in depth in the text. This needs to be verified to see if it is related to the initial VOC composition or oxidant concentration, especially the effect of high ozone experiments (ND1/ND5). Additional discussion is needed.

Indeed, the higher SOA formation observed in ND1 and ND5 is likely linked to the experimental conditions. Both experiments had higher initial OA and VOC levels and also elevated $O_3$ levels compared to the other nighttime-first experiments. So, it appears that the higher availability of precursors, oxidants, and OA all contributed to the higher SOA in these experiments. A brief discussion of this point has been added.

**Reviewer #2**

**(1)** *This manuscript characterizes the atmospheric aging of biomass burning emissions from olive wood burning as well as olive wood mixed with some pine kindling. The focus is on the evolution of OA composition, formation of SOA, and the water-soluble oxidative potential during combined daytime-nighttime aging and nighttime-daytime aging experiments. Overall, the study is very novel and will make an important contribution to our understanding of biomass burning emissions and their evolution in the atmosphere. The data are high quality and robustly support the stated conclusions. The writing is clear and the manuscript is well organized. I highly recommend the manuscript for publication after the following items are addressed:*

We thank the reviewer for the positive assessment of our work and for highlighting the novelty, high quality of the data, and the importance of the contribution. We greatly appreciate the constructive comments provided. Below, we respond (in black) to each point raised (in blue) and outline the corresponding revisions made in the manuscript.

**(2)** *I suggest a major revision of the Conclusions section. This is mostly a restating of key findings already presented in the abstract and results sections. A discussion of the importance, implications, and limitations are mostly missing from this section.*

We have revised the Conclusion section to better highlight the broader importance and implications of our findings, as well as to acknowledge the study's limitations.

**(3)** *For all of the OP results, additional discussion surrounding the water-soluble and water-insoluble fractions should be added. This study only characterizes the water-soluble OP (which is fine), but this is not the total. Studies have demonstrated that BB emissions have insoluble OP, as well. We know that OA generally becomes more water-soluble as it ages/oxidizes. This somewhat complicates interpretation of the OP evolution because the total is not quantified (e.g., what if aging transforms insoluble OP into water-soluble OP, with little impact on total OP? Something like this at least seems plausible).*

Indeed, our study focused only on the water-soluble oxidative potential (WS-OP), as measured by the DTT assay, and therefore does not capture the total OP, which also includes water-insoluble components. We fully agree that this distinction can be important especially when OA becomes more water-soluble during aging. We have added discussion of this point including discussion of previous studies of the changes of water solubility of bbOA during aging. We also mention this limitation in the corresponding section of the Conclusions and highlight the need for future work combining assays targeting the total OP.

**(4)** *Some additional discussion related to the DTT assay is warranted and some of the text should be edited (for example, lines 86-87). While it is true that the DTT assay is widely used because it is relatively simple, fast, and inexpensive, there are problems with the assay and the assumption that it is a surrogate for toxicity. From the Dominutti et al. (2025) AMT article that is cited: "To date, it remains unclear which oxidative potential (OP) assay is most effective at predicting health outcomes related to oxidative stress. Thus, based on current knowledge and epidemiological evidence, two complementary OP assays (a thiol-based probe (OP DTT or*

*OPGSH) and another one (OPOH, OPAA or another) are recommended to provide a better picture of the potential oxidizing damage from PM compounds and to strengthen the power of epidemiological studies. ... Finally, the final choice of the best OP test (or combination) must be based on epidemiological evidence, which has begun to be studied only recently and needs more hindsight to be determined."* To be clear, I am not suggesting the authors go back and add another measure of OP to their study, however, the limitations of having one measure of OP should be clearly stated.

We appreciate the reviewer's insightful comment. We agree that while the DTT assay is widely used due to its relative simplicity, speed, and low cost, it also has its limitations. In the revised manuscript, we have added a statement in the Introduction acknowledging these limitations and emphasizing that the use of a single OP assay cannot be expected to capture the full complexity of aerosol toxicity. We also now cite Dominutti et al. (2025) to highlight the need for complementary assays and epidemiological evidence to strengthen the interpretation of OP measurements.

**(5)** The temperature of all experiments needs to be given in Table 1.

The initial temperatures of all conducted experiments were included in a separate column in Table 1, following the advice of the reviewer.

**(6)** Finally, some discussion of the experimental RH is warranted. This especially may affect interpretation of the nighttime aging experiments because nighttime aging in the atmosphere will often occur under much higher RH conditions, where there will often be significant ALWC. This should be considered when translating the experimental results to atmospheric conditions.

In this study, experiments were conducted at relatively low RH (12–24%) to minimize condensation and sampling artefacts, while enabling controlled comparisons across oxidation scenarios. We acknowledge that atmospheric nighttime aging often occurs at much higher RH, where the aerosol liquid water content can substantially influence multiphase chemistry and SOA formation. Indeed, we have also performed oxidation experiments of the same emissions at higher RH (50–70%) to explicitly investigate these effects, and those results will be presented in a forthcoming companion study. In the present work, we therefore focus on the low-RH conditions, and caution that the findings should be interpreted in the context of this limitation. To address this, we have added a clarifying statement in the Discussion section noting that our results represent low-RH conditions and may differ under more humid atmospheric environments.

**(7)** This is a minor point and perhaps does not even need any changes to the manuscript, but it seems odd that the primary BC concentrations in experiments DN3, DN5, DN6, and ND2 were so low relative to primary OA given the MCEs of 0.96-0.99?

The relatively low primary black carbon (BC) concentrations in experiments DN3, DN5, DN6, and ND2, despite high MCEs of 0.96 – 0.99, likely reflect variability in combustion conditions and fuel composition during our biomass burning experiments. Such variability can lead to differences in OA-to-BC ratios even under high MCE conditions. We note that these differences

do not affect the overall conclusions of the study regarding SOA formation and oxidative potential. We have added a brief note regarding this point in the revised manuscript.

**Reviewer #3**

**(1)** This study investigated how the oxidation sequence (day-to-night and night-to-day) affects biomass burning SOA chemical composition and oxidative potential through controlled chamber experiments simulating realistic diurnal oxidation cycles. Both sequences enhanced organic aerosol (OA) levels by 35-90%. The daytime-first cycle drove rapid, intense daytime oxidation, quickly increasing the O:C ratio. The nighttime-first cycle showed a more gradual, two-step O:C increase to a similar final value of O:C ratio. Spectral evolution also differed initially but resulted in nearly identical aged OA spectra Both cycles effectively reduced precursor compounds like furans and aromatics while increasing aldehydes, ketones, and carboxylic acids. The water-soluble oxidative potential (WS-OP) increased significantly in both cycles. However, the final WS-OP values were statistically similar. Overall, this is a well-designed study, and the characterization and thorough analysis of SOA and its compositions are commendable. However, further analysis and discussion regarding WS-OP are warranted. Once these points are addressed, I would recommend the publication of this work in ACP. Below are more detailed comments and suggestions:

We thank the reviewer for the positive evaluation of our manuscript and for the constructive comments, which have helped us clarify and strengthen the presentation of our work. Below, we provide detailed responses (in black) to the reviewer's comments (in blue) and indicate the corresponding revisions made to the manuscript.

**(2)** It is unclear whether the experimental conditions for all night-to-day (ND) and day-to-night (DN) samples are identical. What are the key differences? While the authors provide some information in Table 1, a summary of these differences would be helpful. More importantly, can the authors clarify whether these differences can explain the variations in SOA formation and OP?

While all night-to-day (ND) and day-to-night (DN) experiments were conducted under the same general initial chamber conditions (temperature, relative humidity, and sampling protocol), there were the unavoidable in these chamber experiments differences in the initial aerosol and gas-phase composition, including the starting organic aerosol (OA) mass, black carbon (BC) content, and oxidant concentrations (e.g., $O_3$, $NO_2$). These differences are summarized in Table 1 and further detailed in the Methods section. These variations can in general explain the observed differences in SOA formation and OP between ND and DN experiments. We have added a summary of these key differences and their implications to the revised manuscript.

**(3)** The discussion on OP results is too simple. Correlation analysis was carried out on WS-OP vs. O:C ratio and aged fractions. More correlation analysis should be done to enhance the findings. For example, OP vs. OC, and all the organics species that were quantified and all compositions in Table 2.

We have followed the advice of the reviewer and quantified the correlations of OP with OC values of Table S3 and the various organic species that were quantified and summarized in Tables 1, 2, and S1. The results are summarized in the revised paper, and the details can be found in the Supplementary Information (Fig. S13-S19).

**(4)** It is very interesting that the intrinsic DTT activity drops in nighttime oxidation after daytime oxidation in the DN case while the DTT activity increases in both oxidation in the ND case. Could the authors provide an explanation for this observation?

These contrasting trends in DTT activity likely reflect differences in the aging of the nighttime chemistry products with OH versus the aging of the daytime chemistry products with $NO_3$. The behavior observed in the DN case could be explained by the production of compounds with lower OP or lower solubility or both when the biomass burning emissions that have already reacted with OH, react with the $NO_3$ radical. On the other hand, the reverse order of reactions appears to lead to products with higher OP or higher solubility or both. This does not appear to be due to the O:C of the products, but rather on their chemical structure. Future work is needed for the identification of these later generation products and the quantification of their OP and water solubility. These potential explanations are discussed in the revised paper.

**(5)** It would be helpful to present Figure 6 to illustrate DTT activity, allowing for a clearer understanding of both absolute and percentage changes in DTT.

We have followed the advice of the reviewer and extended Figure 6 accordingly.

**(6)** The conclusion section would benefit from a more in-depth discussion on the implications of diurnal oxidation cycles on SOA formation and oxidative potential.

We have revised the Conclusions section to provide a more in-depth discussion of how diurnal oxidation cycles influence SOA formation and OP, highlighting the implications for atmospheric chemistry and particle toxicity.